# Uptake of exogenous serine is important to maintain sphingolipid homeostasis in *Saccharomyces cerevisiae*

Bianca M. Esch[1], Sergej Limar[1], André Bogdanowski[2,3], Christos Gournas[4¤], Tushar More[5], Celine Sundag[1], Stefan Walter[6], Jürgen J. Heinisch[7], Christer S. Ejsing[5,8], Bruno André[4], Florian Fröhlich[1,6]*

1 Department of Biology/Chemistry, Molecular Membrane Biology Group, University of Osnabrück, Osnabrück, Germany, 2 Department of Biology/Chemistry, Ecology Group, University of Osnabrück, Osnabrück, Germany, 3 UFZ–Helmholtz Centre for Environmental Research Ltd, Department for Ecological Modelling, Leipzig, Germany, 4 Molecular Physiology of the Cell, Université Libre de Bruxelles (ULB), Institut de Biologie et de Médecine Moléculaires, Gosselies, Belgium, 5 Department of Biochemistry and Molecular Biology, Villum Center for Bioanalytical Sciences, University of Southern Denmark, Odense, Denmark, 6 Center of Cellular Nanoanalytics Osnabrück, Osnabrück, Germany, 7 Department of Biology/Chemistry, Division of Genetics, University of Osnabrück, Osnabrück, Germany, 8 Cell Biology and Biophysics Unit, European Molecular Biology Laboratory, Heidelberg, Germany

¤ Current address: Microbial Molecular Genetics Laboratory, Institute of Biosciences and Applications (IBA), National Centre for Scientific Research "Demokritos" (NCSRD), Agia Paraskevi, Greece
* florian.froehlich@uni-osnabrueck.de

**Data Availability Statement:** All relevant data are within the manuscript and its Supporting Information files.

## Abstract

Sphingolipids are abundant and essential molecules in eukaryotes that have crucial functions as signaling molecules and as membrane components. Sphingolipid biosynthesis starts in the endoplasmic reticulum with the condensation of serine and palmitoyl-CoA. Sphingolipid biosynthesis is highly regulated to maintain sphingolipid homeostasis. Even though, serine is an essential component of the sphingolipid biosynthesis pathway, its role in maintaining sphingolipid homeostasis has not been precisely studied. Here we show that serine uptake is an important factor for the regulation of sphingolipid biosynthesis in *Saccharomyces cerevisiae*. Using genetic experiments, we find the broad-specificity amino acid permease Gnp1 to be important for serine uptake. We confirm these results with serine uptake assays in *gnp1Δ* cells. We further show that uptake of exogenous serine by Gnp1 is important to maintain cellular serine levels and observe a specific connection between serine uptake and the first step of sphingolipid biosynthesis. Using mass spectrometry-based flux analysis, we further observed imported serine as the main source for *de novo* sphingolipid biosynthesis. Our results demonstrate that yeast cells preferentially use the uptake of exogenous serine to regulate sphingolipid biosynthesis. Our study can also be a starting point to analyze the role of serine uptake in mammalian sphingolipid metabolism.

**Funding:** Florian Fröhlich is supported by the DFG grant FR 3647/2-1 and the SFB944 (P20) (https://www.dfg.de/). Sergej Limar and André Bogdanowski are supported by the EvoCell Graduate School of the University of Osnabrück. This research was supported by the VILLUM Foundation (VKR023439, C.S.E. veluxfoundations.dk) and the University of Southern Denmark (SDU2020, C.S.E. www.sdu.dk). The funders had no role in study design, data collection and analysis, decision to publish, or preparation of the manuscript.

**Competing interests:** The authors have declared that no competing interests exist

## Author summary

Sphingolipids (SPs) are membrane lipids globally required for eukaryotic life. In contrast to other lipid classes, SPs cannot be stored in the cell and therefore their levels have to be tightly regulated. Failure to maintain sphingolipid homeostasis can result in pathologies including neurodegeneration, childhood asthma and cancer. However, we are only starting to understand how SP biosynthesis is adjusted according to need. In this study, we use genetic and biochemical methods to show that the uptake of exogenous serine is necessary to maintain SP homeostasis in *Saccharomyces cerevisiae*. Serine is one of the precursors of long chain bases in cells, the first intermediate of SP metabolism. Our results suggest that the uptake of serine is directly coupled to SP biosynthesis at ER-plasma membrane contact sites. Overall, our study identifies serine uptake as a novel regulatory factor of SP homeostasis. While we use yeast as a discovery tool, these results also provide valuable insights into mammalian SP biology especially under pathological conditions.

## Introduction

Sphingolipids (SPs) are essential structural components of membranes and can also act as signaling molecules. SPs constitute up to 20% of the plasma membrane lipids and form tightly packed lipid bilayers together with sterols in the outer layer of the plasma membrane [1]. In contrast to glycerol-phospholipids (GPLs) and sterols, SPs cannot be stored in the cell. Thus, maintaining SP homeostasis is crucial to sustain membrane integrity and trafficking.

SPs are synthesized at the ER by two metabolic branches in yeast. In one branch, the serine palmitoyltransferase (SPT) catalyzes the condensation of serine and palmitoyl-CoA to yield 3-ketodihydrosphingosine. This short-lived intermediate is directly processed to long chain bases (LCBs). In a second branch, palmitoyl-CoA is elongated to up to 24 or 26 carbons chain length, the very long chain fatty acids (VLCFAs). LCBs and VLCFAs are amide linked to form ceramide [2]. In mammalian cells, ceramide transfer protein (CERT) transports ceramides to the Golgi apparatus [3]. A similar mechanism may occur in yeast cells, but the corresponding ceramide transfer protein has yet to be identified. Nvj2 has been suggested as a possible candidate transfer protein in this context [4]. At the yeast Golgi apparatus, ceramides receive various head groups to yield complex SPs. Complex SPs are transported from the Golgi apparatus to reach their destination at the plasma membrane by vesicular transport [5].

SP levels are regulated by post-translational modifications of key enzymes involved in their synthesis. When SP levels at the plasma membrane are low, the target of rapamycin complex 2 (TORC2) is activated and phosphorylates the yeast Akt homologue Ypk1 (reviewed in [6]). This leads to phosphorylation of the Orm proteins, negative regulators of the SPT, and releases the inhibition of SP biosynthesis [7–9]. Other mechanisms include the regulation of ceramide biosynthesis by Ypk kinases [10], the regulation of Pkh kinases [11], and the regulation of VLCFA biosynthesis [12,13]. Phospho-proteomic studies of SP homeostasis further suggested that the complex SP biosynthetic enzymes are also subject to regulation [14].

In fact, SP metabolism in yeast is regulated at several steps. However, little is known about the role of serine, one of the two substrates of the first and rate-limiting step in this pathway. Serine can be synthesized via the 3-phospho-glycerate pathway [15] or converted from glycine in a reversible reaction catalyzed by the serine hydroxymethyltransferases Shm1 and Shm2 [16]. Additionally, serine can be imported from the medium by plasma membrane permeases, as suggested by overexpression of the encoding genes [17]. Which portion of the different serine pools are incorporated into SPs is unknown. However, serine uptake and flux into the SP

biosynthesis pathway has been shown to increase upon heat stress [18]. Interestingly, the catabolic serine deaminase Cha1 is upregulated by exogenous serine, and LCBs may serve as sensors of serine availability and mediate up-regulation of Cha1 in this response [19]. This indicates a regulatory control between exogenous serine and SP metabolism.

We therefore investigated the role of serine as an additional regulatory factor for SP homeostasis. We analyzed the impact of Gnp1 and Agp1, members of the yeast amino acid transporter (YAT) family [20], on serine uptake and intracellular amino acid concentrations and studied their role in SP homeostasis. Both proteins are dispensable for SP homeostasis under standard growth conditions. However, when SP metabolism is challenged by inhibition of the first steps in the synthetic pathway, Gnp1-dependent serine uptake becomes essential. Using mass spectrometry-based flux analysis we demonstrated the direct integration of imported serine into SPs and found extracellular serine to be the main source for *de novo* SP biosynthesis. Combined, these data reveal an additional, previously unknown, regulatory mechanism to maintain cellular SP homeostasis.

## Results

### Serine is required for SP homeostasis

SP biosynthesis starts with the condensation of serine and palmitoyl-CoA by the rate limiting enzyme SPT. Serine can be synthesized from 3-phospho-glycerate via reactions catalyzed by Ser3/Ser33, Ser1 and Ser2 (Fig 1A) or from glycine by the action of the serine hydroxymethyltransferases Shm1 and Shm2 [16]. To assess the effects of altered serine availability we first tested whether deletion mutants of the 3-phospho-glycerate pathway are auxotroph for serine under different conditions. For this purpose, *ser1Δ*, *ser2Δ*, *ser3Δ* and *ser33Δ* mutants were spotted onto the respective drop-out media. When *SER1* or *SER2* were deleted, cells did not grow on medium lacking serine, while single mutants in the redundant gene pair (*ser3Δ* or *ser33Δ*) were able to grow (Fig 1B). This confirms the previously observed serine auxotrophy for a single *SER1* or *SER2* deletion [21].

In a genome wide chemical genetic screen we had previously identified a *ser2Δ* strain as sensitive to the depletion of SPs by myriocin [22]. Myriocin is a suicide inhibitor of the SPT, consisting of the subunits Lcb1, Lcb2 and the regulatory subunit Tsc3 (Fig 1C) [23]. We therefore tested growth of a wild-type (WT) strain and the serine biosynthesis mutants *ser1Δ*, *ser2Δ*, *ser3Δ* and *ser33Δ* on control plates, plates lacking serine, plates containing myriocin and plates containing myriocin and lacking serine. As shown in Fig 1D, *ser3Δ* and *ser33Δ* cells showed no increased sensitivity to myriocin. In contrast, *ser1Δ* and *ser2Δ* cells were highly sensitive to chemical depletion of SPs by myriocin, demonstrating the importance of serine availability for the maintenance of SP homeostasis. In addition, media without serine renders WT cells more sensitive to chemical depletion of SPs (Fig 1D), suggesting that the presence of exogenous serine is also an important factor to maintain SP homeostasis.

### *GNP1* genetically interacts with serine metabolic genes in yeast

Due to the essential nature of serine for metabolism, one would expect that a defect in a major serine permease would be lethal for a serine auxotroph strain. In contrast, impairment of either serine uptake or serine biosynthesis should be tolerable under growth conditions where serine is available (Fig 2A). We therefore specifically screened high throughput genetic interaction data for interactions with a *ser2Δ* deletion [24]. We plotted the genetic interaction score of *ser2Δ* (epsilon score) with each gene deletion against the significance of the genetic interaction (Fig 2B; note that similar results were obtained for *ser1Δ*; S1A Fig). Interestingly, the strongest and most significant genetic interaction of the *SER2* gene was observed with two ORFs, *GNP1*

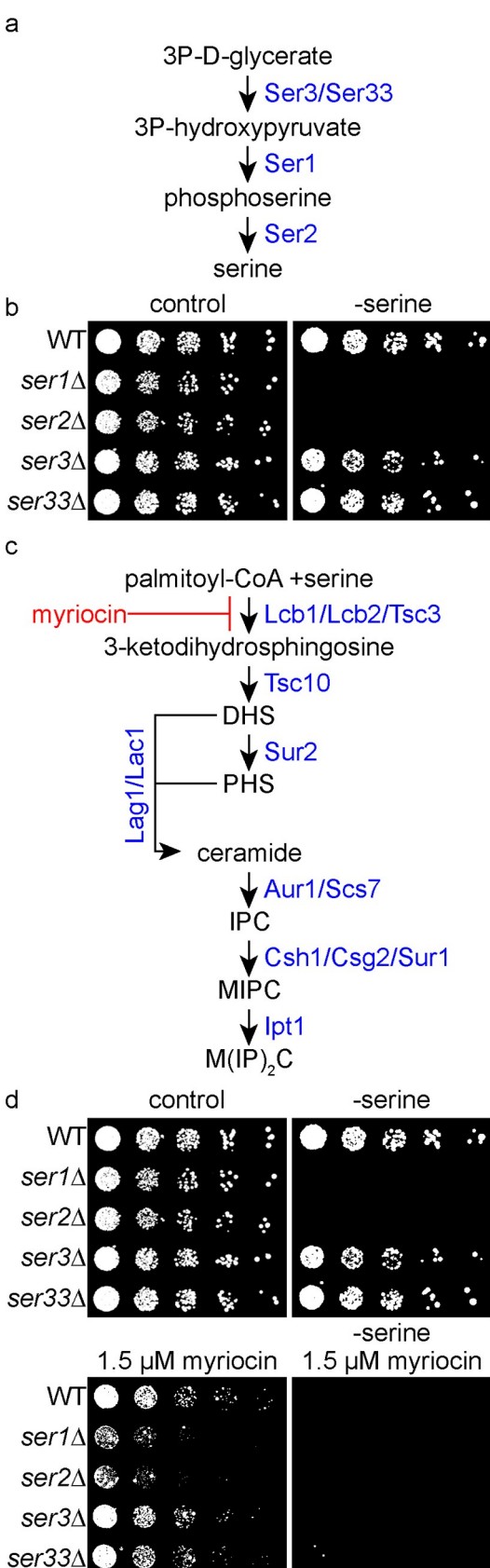

**Fig 1. Serine metabolism is essential for sphingolipid homeostasis in yeast.** (**a**) Model of serine metabolism via the 3-phospho-glycerate pathway. (**b**) Serial dilutions of knockouts of serine biosynthesis genes in the presence (control) or absence of serine (- serine) on synthetic medium (SDC). The strains used are from top to bottom: wild-type (WT), *ser1Δ, ser2Δ, ser3Δ* and *ser33Δ*. (**c**) Model of sphingolipid (SP) metabolism in yeast. Myriocin (red) is an inhibitor of the SPT (Lcb1, Lcb2, Tsc3). (**d**) Exogenous serine is essential for survival under SP depleted conditions. Serial dilutions of WT, *ser1Δ, ser2Δ, ser3Δ* and *ser33Δ* cells on SDC medium. Control plates (upper left), plates without serine (upper right panel), plates containing 1.5 μM myriocin (lower right) and plates containing 1.5 μM myriocin with no serine available (lower right) are displayed.

and *YDR509W*, with the latter being annotated as a dubious open reading frame partially overlapping with *GNP1*.

Other significant hits were part of the TORC1 signaling pathway (S1B Fig). In line with our hypothesis, *GNP1* encodes a broad-specificity amino acid permease that was previously suggested to be necessary for the uptake of serine and other amino acids [17].

We proceeded by studying the genetic interaction of *SER2* and *GNP1* using tetrad analyses. As expected, the double knockout of *ser2Δgnp1Δ* showed a very strong negative genetic interaction as reflected by slow growth and very small colonies (Fig 2C). *GNP1* has a paralog, *AGP1*, which was presumably generated by the whole genome duplication in *S. cerevisiae* [25]. The respective epistasis analysis with the *ser2Δ* mutant did not reveal any synthetic phenotype (Fig 2D), suggesting that Gnp1 is the major permease with regard to serine uptake. The fact that the triple knockout *ser2Δgnp1Δagp1Δ* mutants did not produce any viable progeny indicates a complete lack of serine import in the absence of both permeases, Agp1 and Gnp1 (Fig 2E).

## Gnp1 is the major serine transporter in yeast cells

Our genetic analyses suggested Gnp1 to be the major serine permease in yeast with only minor contributions of its paralog Agp1. This was confirmed by determination of serine uptake kinetics with radioactively labelled serine in *gnp1Δ*, *agp1Δ* and *gnp1Δagp1Δ* mutants as compared to WT cells (Fig 3A). As expected, the *gnp1Δ* strain showed a strong reduction (65%) in serine uptake compared to WT cells already 5 minutes after adding the labelled substrate (Fig 3A). Similarly, *agp1Δ* cells displayed a pronounced effect on serine uptake although substantially weaker than *gnp1Δ* cells with about 34% decrease in serine uptake compared to WT cells after 5 minutes. This observation is not reflected in our genetic analyses where Gnp1 appears to be responsible for most of the serine uptake and potentially represents a difference in the serine availability of the respective growth media. A strain lacking both, *GNP1* and *AGP1*, showed very low transport activity (Fig 3A). These results confirm that Gnp1 is the major serine transporter in yeast under the used growth conditions.

Serine is a metabolic precursor in multiple pathways besides SP biosynthesis. Thus, it is a donor of one-carbon units to folate [26], important for the synthesis of GPLs such as phosphatidylserine [27,28] and phosphatidylethanolamine [29], and it is required for protein synthesis (Fig 3B). To test the effect of a *GNP1* deletion on protein biosynthesis, we grew WT cells, *gnp1Δ* cells, *agp1Δ* cells, *gnp1Δagp1Δ* cells and *ser2Δ* cells in the presence of $[^{13}C_3{}^{15}N_1]$-serine and measured its incorporation into proteins using mass spectrometry-based proteomics. Extracted proteins were digested with the α-Lytic protease and only peptides containing exactly one serine were analyzed. In WT cells 93.8% of the analyzed peptides contained a $[^{13}C_3{}^{15}N_1]$-serine (Fig 3C). The serine auxotroph *ser2Δ* reached an incorporation rate of 92.7%. In line with the data reported above, the $[^{13}C_3{}^{15}N_1]$-serine incorporation in *gnp1Δ* cells was reduced to 59% (Fig 3C). In *agp1Δ* cells the serine incorporation was comparable to WT cells with an incorporation rate of 90.8%. The double deletion of *GNP1* and *AGP1* resulted in an 81% decrease of serine incorporation into proteins compared to WT cells. It is important to

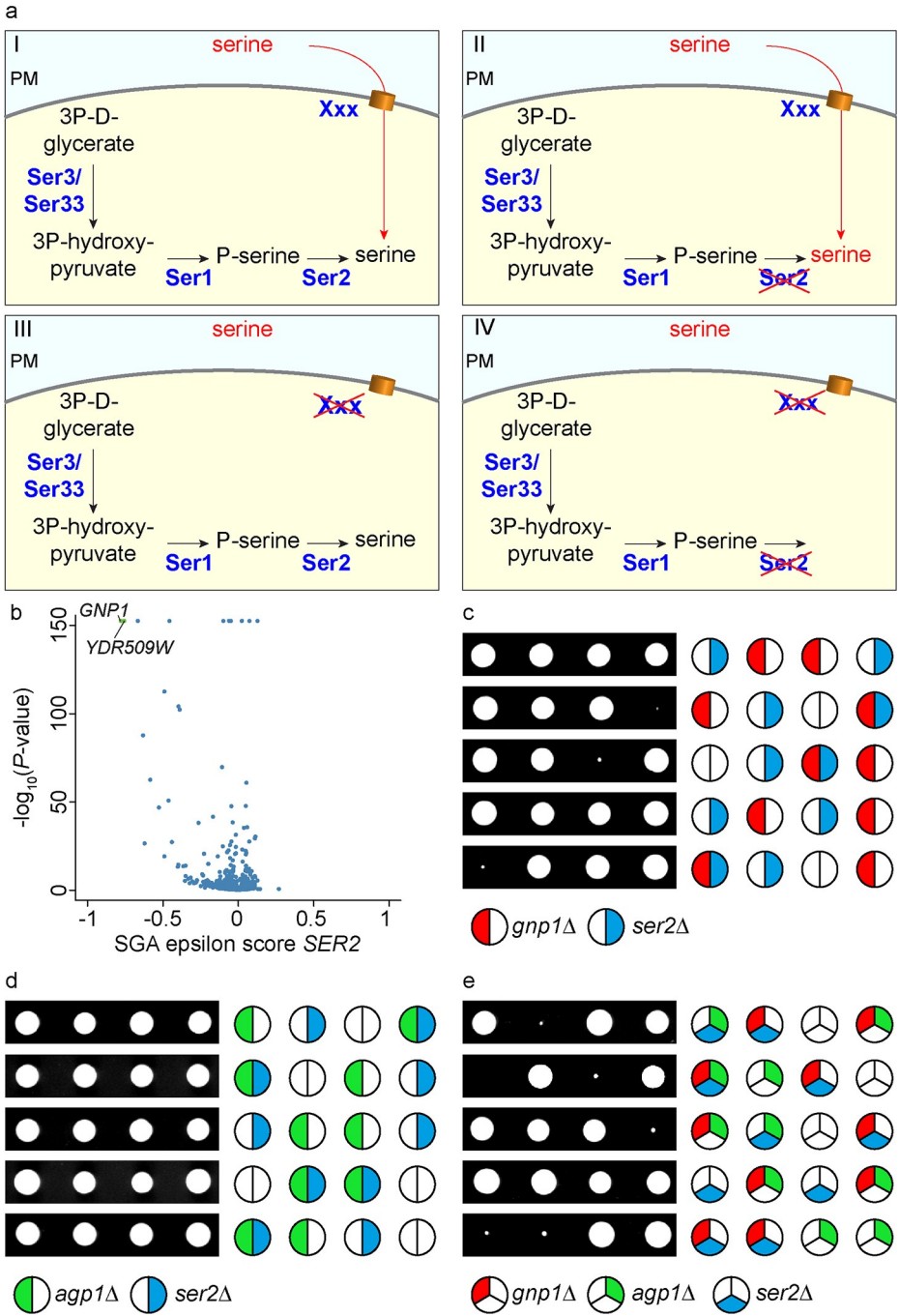

**Fig 2. GNP1 genetically interacts with serine metabolism. (a)** Outline for the identification of the plasma membrane serine transporter in yeast. WT cells (I, upper left) as well as *SER2* knockout cells (II, upper right) and knockouts of the serine transporter in yeast (gene XXX, III lower left) should grow fine. Deletion of both, *SER2* and the serine transporter should result in synthetic lethality (IV, lower right). **(b)** The genetic interaction score (epsilon score) of *SER2* is plotted against the negative LOG$_{10}$ of the P-value of the interactions. The volcano plot shows significant negative genetic interactions on the left side of the plot. Highly significant interactions are shown in green. Data are taken from (25). **(c)** Tetrad analysis of *gnp1Δ* (red) mutants crossed with *ser2Δ* (blue). **(d)** Tetrad analysis of *agp1Δ* (green) mutants crossed with *ser2Δ* (blue). **(e)** Tetrad analysis of *ser2Δ* (blue) mutants crossed with *gnp1Δagp1Δ* (red and green, respectively).

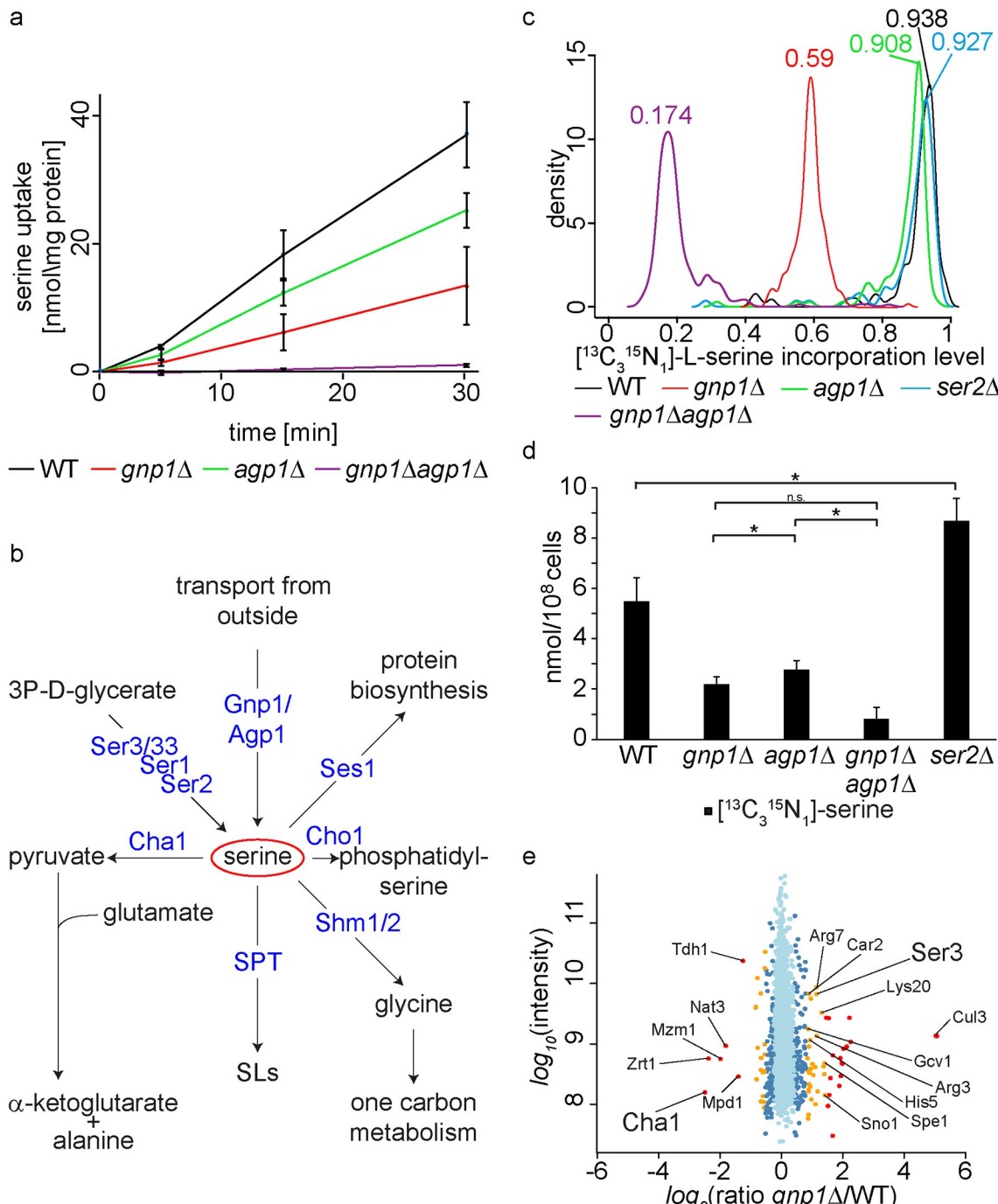

**Fig 3. Gnp1 and Agp1 are essential for the uptake of exogenous serine. (a)** WT cells (black line), *gnp1Δ* cells (red line), *agp1Δ* cells (green line) and *gnp1Δagp1Δ* cells (purple line) were grown in SDC medium. Cells were washed and incubated with [14C]-serine. The uptake rate of [14C]-serine was measured. Error bars correspond to standard deviations. n = 3. **(b)** Schematic outline of the main serine consuming and producing processes. **(c)** WT cells (black line), *gnp1Δ* cells (red line), *agp1Δ* cells (green line), *gnp1Δagp1Δ* cells (purple line) and *ser2Δ* cells (blue line) were grown in the presence of stable isotope labelled, [$^{13}$C$_3$$^{15}$N$_1$]-serine in SDC medium. Graphs represent the density function of the rates of [$^{13}$C$_3$$^{15}$N$_1$]-serine incorporation into all peptides containing a single serine. **(d)** Cellular concentrations of free [$^{13}$C$_3$$^{15}$N$_1$]-serine in WT, *gnp1Δ* cells, *apg1Δ* cells, *gnp1Δagp1Δ* cells and *ser2Δ* cells grown in SDC medium with [$^{13}$C$_3$$^{15}$N$_1$]-serine were analyzed by mass spectrometry. Error bars represent standard deviations. n = 3. *, P-value <0.05, calculated from t-test. **(e)** Proteome comparison of lysine labeled WT cells and [$^{13}$C$_6$$^{15}$N$_2$]-lysine labeled *gnp1Δ* cells grown in SILAC medium. Protein intensities are plotted against normalized SILAC ratios of heavy (*gnp1Δ*) to light (WT). Significant outliers are colored in red (p < 1e$^{-11}$), orange (p<1e$^{-4}$) or dark blue (p < 0.05); other proteins are shown in light blue.

note that direct serine uptake was measured in minutes (Fig 3A). Here, incorporation of serine into proteins was measured after long term growth in the presence of $[^{13}C_3{}^{15}N_1]$-serine.

To have comparable uptake results we also measured free $[^{13}C_3{}^{15}N_1]$-serine levels in the cells by mass spectrometry. These results again show that less serine is taken up in *gnp1Δ* cells, *agp1Δ* cells and *gnp1Δagp1Δ* cells compared to WT cells (Fig 3D). Interestingly, the *ser2Δ* strain had more free $[^{13}C_3{}^{15}N_1]$-serine compared to WT cells, suggesting increased uptake (Fig 3D). Again, these results confirm that Gnp1 is the major serine transporter in yeast with smaller contributions of its paralog Agp1. However, in the absence of *GNP1* and *AGP1* there are still low amounts of serine taken up, but these levels are not sufficient to allow for growth of the *gnp1Δagp1Δser2Δ* triple mutant (Fig 2E).

In the course of proteomics analyses of *gnp1Δ* cells we gained further evidence that Gnp1 directly contributes to intracellular serine levels. We measured the entire proteome of *gnp1Δ* cells compared to WT cells using stable isotope labelling by amino acids in cell culture (SILAC) [30] combined with mass spectrometry-based proteomics. Among the 2322 proteins quantified (S5 Source Data Fig 3E) we found an up-regulation of several metabolic enzymes, with a prevalence in amino acid metabolic processes (according to GO term analysis). Amongst these enzymes are the 3-phosphoglycerate dehydrogenase Ser3 catalyzing a rate limiting step in serine and glycine metabolism, as well as a subunit of the glycine decarboxylase Gcv1 (Fig 3E). Interestingly, the most down-regulated protein was the catabolic serine deaminase Cha1 (Fig 3E). Cha1 is under strong transcriptional control of the Cha4 transcription factor and is known to be upregulated by exogenous serine [19]. Another protein that is slightly but significantly downregulated (p value = 0.0078, normalized ratio = 0.78) is the negative regulator of the serine palmitoyl-transferase Orm2 [7,31]. In addition, several genes involved in ergosterol biosynthesis were significantly downregulated (Erg11, Erg24, Hmg1, Erg1, Are2, Erg26). Together, this indicates that intracellular serine levels in *gnp1Δ* cells are decreased, and thus forces cells to adjust by reprogramming their metabolism.

## Intracellular serine concentrations depend on serine uptake

To predict how serine biosynthesis and uptake are correlated, we used flux variability analysis (FVA) which allows a prediction of possible fluxes through a reconstructed metabolic network. First, we analyzed the contribution of cellular processes involving serine (Fig 4A). Our results highlighted the glycine hydroxymethyltransferases Shm1 and Shm2 as two main producers and consumers of serine within the cell, respectively (Fig 4A). Additionally, serine synthesis by the phosphoserine phosphatase Ser2 and uptake of external serine were identified as potential serine sources. To establish how serine uptake is determined by the relevant serine fluxes, we modelled the net flux through Ser2, Shm1 and Shm2 at varied serine uptake rates. According to the model, serine synthesis outweighs serine consumption only at low serine uptake rates (Fig 4B). At higher serine uptake rates, excess serine appears to be converted to glycine by Shm2. The flux prediction for the sum of Shm1 and Shm2 fluxes was not changed compared to the flux prediction including Ser2 (S2A Fig). Thus, the impact of the 3-phospho-glycerate pathway seems to be very low regarding total serine fluxes.

To test this model, we measured cellular amino acid concentrations by mass spectrometry. We used media without amino acids (except for serine where indicated) together with prototrophic WT and *gnp1Δ* cells. The amount of detected serine in WT cells was significantly increased by 67% in media containing serine compared to cells grown in the absence of serine. This indicates a pronounced effect of serine uptake on intracellular serine concentrations (Fig 4C). In contrast, serine levels in *gnp1Δ* cells with and without serine were comparable to WT cells grown in the absence of serine. This suggests that intracellular serine levels are directly

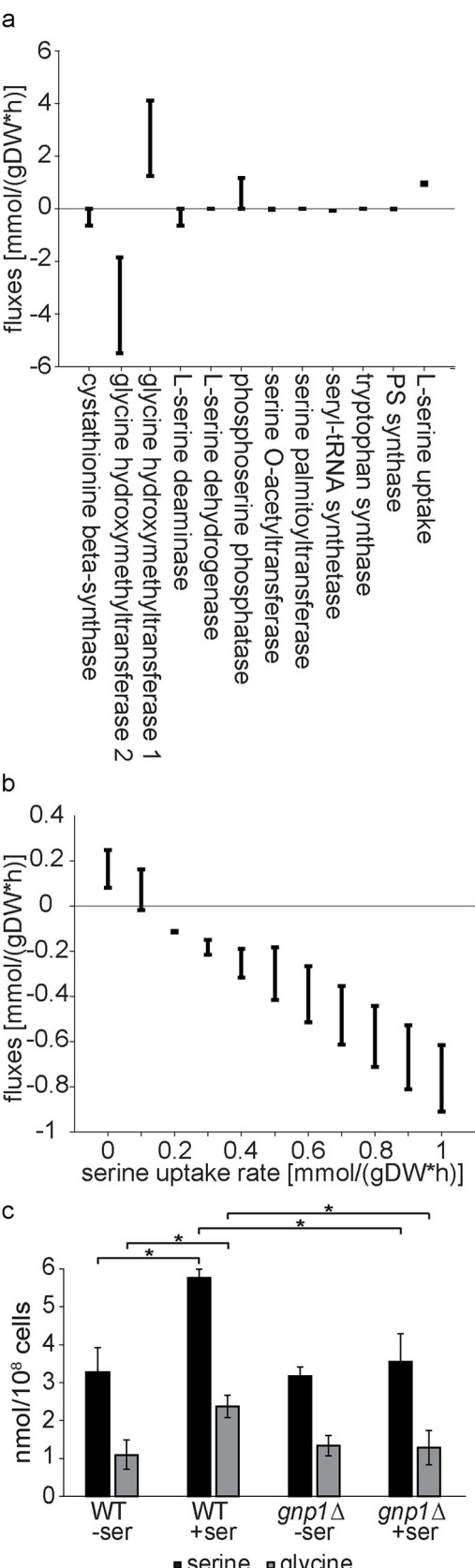

**Fig 4. Intracellular serine concentrations are dependent on serine uptake. (a)** Predicted fluxes of serine. Flux variability of metabolic reactions, which produce or consume serine, as predicted by flux variability analysis (FVA). Positive and negative fluxes correspond to serine production and consumption, respectively. Fluxes are represented in mmol per gram dry weight per hour. **(b)** Predicted net flux of serine producing reactions at varying serine uptake rates. Variability of net flux through Shm1, Shm2 and Ser2 at varying serine uptake rates, as predicted by FVA. Positive and negative fluxes correspond to serine production and consumption, respectively. Fluxes and serine uptake rates are represented in mmol per gram dry weight per hour. **(c)** Cellular serine and glycine concentrations. Prototroph WT and *gnp1Δ* cells were grown in synthetic medium without amino acids and with and without serine. Serine and glycine concentrations from whole cell lysates were analyzed by mass spectrometry. Error bars represent standard deviations. n = 3. *, P-value <0.05, calculated from t-test. Differences between data that are not labelled with a star are not significant.

depending on Gnp1 mediated serine uptake (Fig 4C). Lysine levels were constant with and without serine in WT and in *gnp1Δ* cells (S2B Fig), indicating no changes in unrelated amino acids. In contrast, glycine levels showed comparable changes as serine levels. Glycine levels were 120% increased for WT cells grown in media with serine compared to WT cells grown without serine (Fig 4C). In addition, glycine levels also did not increase in *gnp1Δ* cells grown in the presence of serine. As there was no glycine in the media, this indicates conversion of the transported serine to glycine as predicted by the FVA.

## Serine uptake by Gnp1 is important to maintain SP homeostasis

To test if serine availability is important to maintain SP homeostasis, we next analyzed the contribution of Gnp1-dependent serine uptake on SP metabolism. First, serial dilutions of WT, *gnp1Δ*, *agp1Δ* and *gnp1Δagp1Δ* cells were spotted on different drop-out media in the presence or absence of serine and myriocin. As expected, all strains grew well on all plates in the absence of myriocin (Fig 5A). This also rules out any important role of Gnp1 and Agp1 in serine biosynthesis. Interestingly, *gnp1Δ* cells showed strong growth defects after inhibition of SP biosynthesis by myriocin, even under conditions were growth of WT cells was not affected (Fig 5A). Thus, Gnp1-mediated serine uptake is essential under conditions of impaired SP metabolism. In line with our genetic analyses and serine uptake assays, growth of *agp1Δ* cells was not affected by the presence of myriocin, while an additive effect was observed in the *gnp1Δagp1Δ* double mutant (Fig 5A). Confirming the initial results, all strains were highly sensitive to myriocin in the absence of external serine (Fig 5A).

To exclude possible side effects of myriocin, we tested if the effect of myriocin can be recovered by the addition of phytosphingosine (PHS). PHS, together with dihydrosphingosine (DHS), is one of the two LCBs in yeast. Myriocin inhibits the synthesis of 3-ketodihydrosphingosine, the short-lived precursor of LCBs (Fig 1C). We used serial dilutions to test the effect of myriocin and PHS on WT, *gnp1Δ*, *agp1Δ* and *gnp1Δagp1Δ* cells. We found that addition of 25 μM PHS recovered the growth deficiency of *gnp1Δ* cells at 0.5 μM myriocin (Fig 5B), suggesting that there are no side effects of myriocin treatment.

Gnp1 is a general amino acid permease which was previously described to transport glutamine, leucine, threonine, cysteine, methionine and asparagine in addition to serine [17]. For this reason, we tested if the transport of the other amino acids affects the connection between Gnp1 and SP metabolism. We performed serial dilutions of prototrophic WT, *gnp1Δ*, *agp1Δ* and *gnp1Δagp1Δ* cells on media without amino acids (except of serine were indicated). Without serine in the media, *gnp1Δ* cells showed no increased sensitivity to myriocin compared to WT cells (Fig 5C). This indicated that the sensitivity of *gnp1Δ* cells to myriocin is induced by amino acids in the media. On plates containing serine and 1.5 μM myriocin *gnp1Δ* cells were slightly decreased in growth compared to WT cells (Fig 5C). However, this effect was not as pronounced as observed before (Fig 5A). The *gnp1Δagp1Δ* cells were highly decreased in

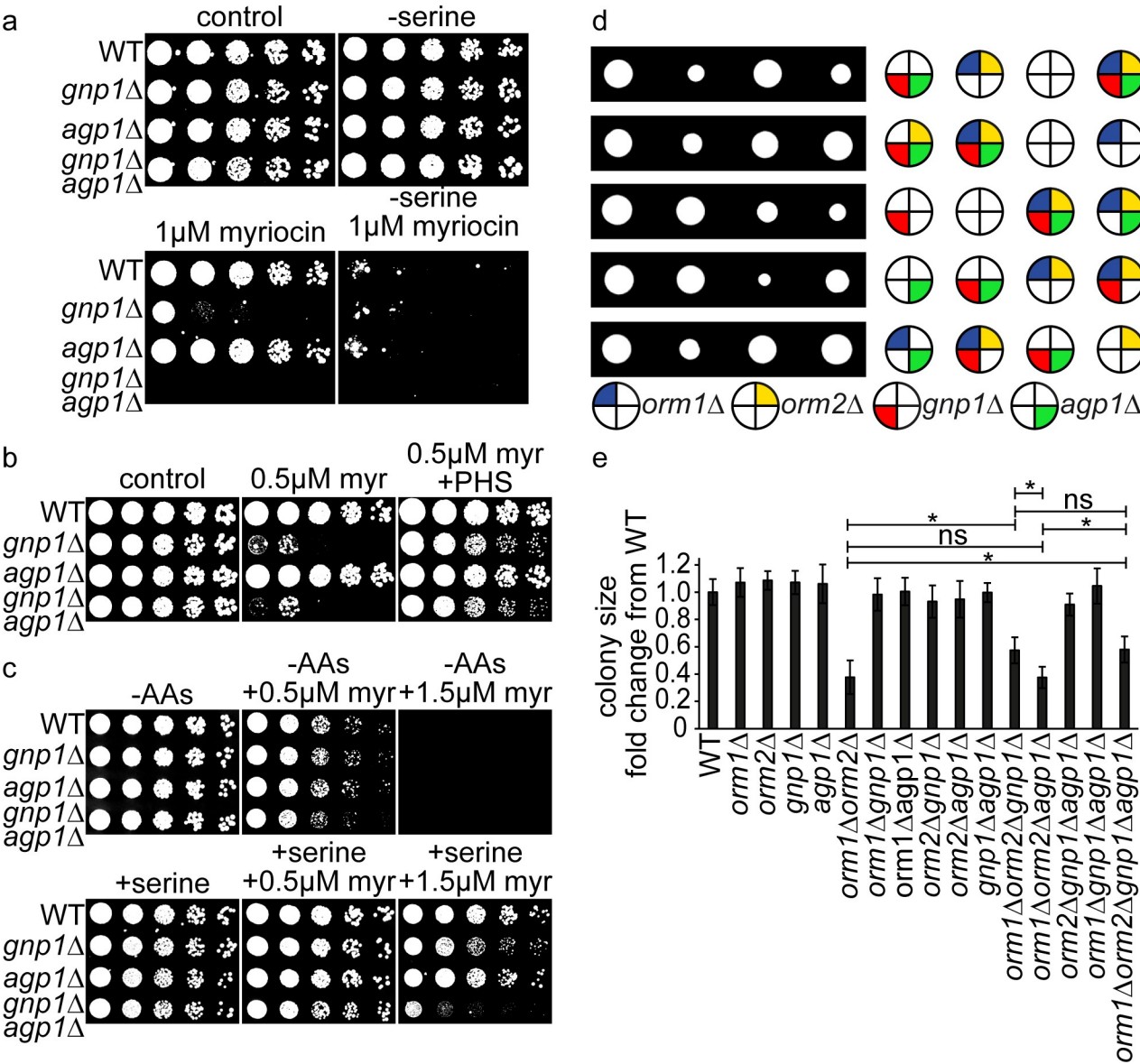

**Fig 5. Gnp1 dependent serine uptake is essential to maintain sphingolipid homeostasis. (a)** Serial dilutions of WT, *gnp1Δ* cells, *agp1Δ* cells and *gnp1Δagp1Δ* cells on synthetic medium. Control plates (upper left), plates without serine (upper right), plates containing 1 μM myriocin (lower left) and plates containing 1 μM myriocin with no serine available (lower right) are displayed. **(b)** Serial dilutions of WT, *gnp1Δ* cells, *agp1Δ* cells and *gnp1Δagp1Δ* cells on YPD medium. Control plates (left), plates containing 0.5 μM myriocin (middle) and plates containing 0.5 μM myriocin and 25 μM phytosphingosine (PHS) (right) are displayed. **(c)** Serial dilutions of prototroph WT, *gnp1Δ* cells, *agp1Δ* cells and *gnp1Δagp1Δ* cells on plates with synthetic medium without amino acids (upper row) and plates without amino acids (-AAs) with serine (lower row), each with 0.5 μM (middle) and 1.5 μM myriocin (right). **(d)** Tetrad analysis of *orm1Δorm2Δ* cells (dark blue and yellow, respectively) crossed with *gnp1Δ* (red) cells and followed by the deletion of *AGP1* (green). **(e)** Quantification of tetrad analyzis shown in (d). Relative colony sizes of 35 tetrades of diploid *orm1Δorm2Δgnp1Δagp1Δ* cells shown as fold change from WT tetrades. Error bars represent standard deviations. *, P-value <0.05, calculated from t-test.

growth on plates with serine and 1.5 μM myriocin (Fig 5C). Together, this indicates the presence of serine as the cause of sensitivity for *gnp1Δ* and *gnp1Δagp1Δ* cells to SP depletion and suggests no contribution of other transported amino acids. The expression of Gnp1 and Agp1 is controlled by the SPS sensor system and increases after addition of amino acids [32]. The difference in the sensitivity of *gnp1Δ* cells towards myriocin in media with amino acids (Fig

5A) and media without amino acids, except of serine, (Fig 5C) could be a result of different expression levels of Gnp1 and Agp1. To test this hypothesis, we tagged Gnp1 and Agp1 c-terminally with the ALFA-tag at their genomic locus [33,34] and analyzed their expression levels in different media (S3C Fig). The functionality of Gnp1-ALFA was verified by genetic interactions with *ser2Δ* in tetrad dissection (S3A and S3B Fig). In YPD and SDC medium Gnp1 was higher expressed then Agp1, while both transporters showed no detectable expression in media without amino acids. The addition of serine increased the expression of both transporters, with higher levels of Agp1 than Gnp1 (S3C Fig). These observations explain changes in myriocin sensitivity upon growth in different media (Fig 5A and 5C). Nevertheless, the *agp1Δ* strain showed no sensitivity to myriocin in both experiments (Fig 5A and 5C). Additionally, the general sensitivity of the cells to myriocin was highly increased in the media without amino acids, consistent with our previous observations that the absence of serine increased the sensitivity towards SP depletion (Fig 5C).

In the next step, we asked if the observed connection between serine uptake and SP homeostasis results in other genetic interactions. As Lcb1 and Lcb2, the two catalytic subunits of the SPT are essential; we used deletions of *ORM1* and *ORM2* as a read-out for genetic interactions of this step of SP biosynthesis. Orm1 and Orm2 are negative regulators of the SPT and release their inhibition after phosphorylation. Moreover, *orm1Δorm2Δ* strains were shown to have increased amounts of LCBs [7]. A *gnp1Δagp1Δ* strain mated with an *orm1Δorm2Δ* strain was therefore subjected to tetrad analysis. The *orm1Δorm2Δ* double mutants showed a growth defect, which was partly restored by the additional deletion of *GNP1*, but not by that of *AGP1*. The *orm1Δorm2Δgnp1Δagp1Δ* quadruple mutants showed the same growth defect as the triple mutants of *orm1Δorm2Δgnp1Δ*, again in line with Gnp1 being the major serine transporter under the tested conditions (Fig 5D and 5E). The recovery of the growth defect by the deletion of *GNP1* indicates a rescue of increased SP levels due to decreased serine uptake. Together, this suggests a regulatory effect of serine uptake on SP metabolism.

To exclude a general effect of diminished serine uptake towards the SP biosynthesis pathway, we performed genetic interaction studies with additional enzymes of the SP biosynthesis pathway. None of the tested gene deletions (*sur2Δ*, *sur4Δ*, *lcb3Δ*, *lcb4Δ*, *scs7Δ*; see Fig 1C for an overview of the SP biosynthesis pathway) displayed a synthetic phenotype with *gnp1Δ* or *ser2Δ* (S4 Fig). This indicates a specific genetic interaction of Gnp1 with the first, serine-consuming step of SP biosynthesis.

To further substantiate these findings, we performed serial dilutions of WT, *gnp1Δ*, *agp1Δ* and *gnp1Δagp1Δ* cells on media containing the inhibitor Aureobasidin A, which blocks the formation of inositol-phosphorylceramides (IPCs). All tested strains showed comparable sensitivity to Aureobasidin A, indicating no interaction of serine uptake with this step of SP biosynthesis (S5A Fig). We also monitored the effect of fatty acid availability with Gnp1-mediated serine uptake by spotting WT, *gnp1Δ*, *agp1Δ* and *gnp1Δagp1Δ* cells on plates containing cerulenin, an inhibitor of the fatty acid synthase. All tested strains (WT, *gnp1Δ*, *agp1Δ* and *gnp1Δagp1Δ*) showed comparable growth defects, ruling out an effect of Gnp1 on palmitoyl-CoA availability (S5B Fig). All these results are in line with our genetic studies, indicating a prominent interaction of the first SP catalyzing step and serine uptake.

## Gnp1-transported serine is the main source for sphingolipid biosynthesis

Finally, we decided to analyze the direct effect of serine uptake on yeast LCB levels as a readout of SPT activity. FVA predicted a direct relationship between increased SPT flux and increased serine uptake (Fig 6A). To experimentally test this connection, we measured PHS levels of WT and *gnp1Δ* cells under high and low concentrations of myriocin by LC-MS. It was previously shown

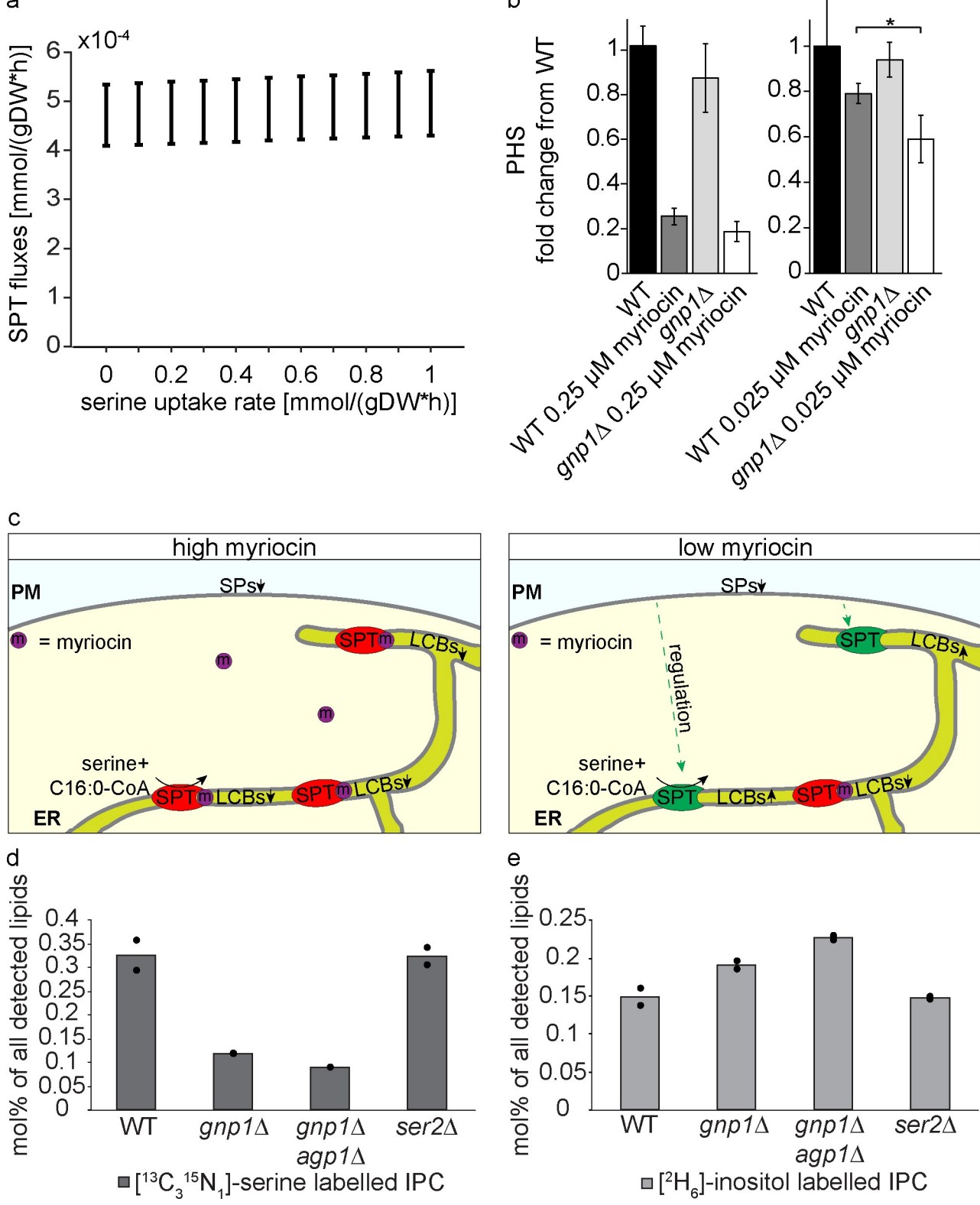

**Fig 6. Imported serine is the main source for SP biosynthesis. (a).** Predicted flux of SPT at varying serine uptake rates in YPD medium. Variability of flux through SPT at varying serine uptake rates, as predicted by FVA. Positive and negative fluxes correspond to serine production and consumption, respectively. SPT fluxes and serine uptake rates are represented in mmol per gram dry weight per hour. **(b)** GNP1 is essential to maintain SP

homeostasis. Mass spectrometric analysis of phytosphingosine (PHS) concentrations in WT and *gnp1Δ* cells grown in the presence or absence of 0.25 μM myriocin or 0.025 μM myriocin in YPD medium. Error bars represent standard deviations. n = 3. *, P-value <0.05, calculated from t-test. **(c)** Model of SPT activity under high and low concentrations of myriocin. Under high concentrations of myriocin (left panel) most SPT complexes are inhibited by myriocin and de novo SP biosynthesis is significantly impaired resulting in decreased SP concentrations. Under low concentrations of myriocin (right panel), only few SPT complexes are inhibited by the suicide inhibitor myriocin while the cell regulates against the decreasing SP levels resulting in the upregulation of the remaining SPT molecules and constant SP concentrations. **(d)** Integration of $[^{13}C_3^{15}N_1]$-serine into inositol-phosphorylceramides (IPC). Cells were labelled with $[^{13}C_3^{15}N_1]$-serine and $[^2H_6]$-inositol over 90 minutes in YPD medium. Lipids were extracted and analyzed via mass spectrometry. Displayed are the amounts of $[^{13}C_3^{15}N_1]$-serine labelled IPCs of WT cells, *gnp1Δ* cells, *gnp1Δagp1Δ* cells and *ser2Δ* cells in mol% per all detected lipids. The average is displayed in bars. Dots correspond to the values of two independent experiments. **(e)** Integration of $[^2H_6]$-inositol into IPCs. Displayed are the amounts of $[^2H_6]$-inositol labelled IPCs of WT cells, *gnp1Δ* cells, *gnp1Δagp1Δ* cells and *ser2Δ* cells in mol% per all detected lipids. The average is displayed in bars. Dots correspond to the values of two independent experiments.

that yeast cells can maintain constant levels of LCBs in the presence of low myriocin concentrations [7]. This was attributed to a negative feedback loop involving TORC2, Slm1/2 proteins, the Ypk kinases and the Orm proteins as negative regulators of the SPT [8,9,35]. At low myriocin concentrations, the SPT is partially inhibited, while the remaining SPT molecules are most likely upregulated by the described feedback mechanism, resulting in the tolerance of WT cells to low concentrations of myriocin (Fig 6C). Our analysis revealed that the *GNP1* deletion had no significant effect on PHS levels as compared to WT cells (Fig 6B). As expected, treatment with a high concentration of 100 ng myriocin per ml (0.25 μM) resulted in a strong depletion of PHS in WT and *gnp1Δ* cells (Fig 6B), suggesting that the cells are not able to compensate the inhibition of SP biosynthesis. However, treatment of the cells with 10 ng myriocin per ml (0.025 μM) resulted in only a small, non-significant decrease in PHS levels in WT cells confirming previous studies [7]. In contrast, PHS levels in *gnp1Δ* cells decreased significantly compared to equally treated WT cells (Fig 6B), suggesting a regulatory impact of serine uptake on SP metabolism.

To establish the correlation between extracellular serine uptake and SP metabolism more directly, we measured the incorporation of $[^{13}C_3^{15}N_1]$-serine and $[^2H_6]$-inositol into SPs. We labelled WT, *gnp1Δ*, *gnp1Δagp1Δ* and *ser2Δ* cells in duplicates for 90 minutes with $[^{13}C_3^{15}N_1]$-serine and $[^2H_6]$-inositol and measured their incorporation into the lipidome by high-resolution shotgun lipidomics. The analysis of $[^{13}C_3^{15}N_1]$-serine labelled IPCs revealed a 64% decrease in serine labelling in *gnp1Δ* cells and a decrease of 73% in *gnp1Δagp1Δ* cells compared to WT cells (Fig 6D). This again, suggests small amounts of serine being taken up even in the absence of Gnp1 and Agp1. However, similar effects were observed in the detected ceramides where no serine labelled ceramides could be identified at all (S6 Fig). The decreased serine labelling of SPs without Gnp1 and the low difference in labelling between the *GNP1* deletion and the *GNP1 AGP1* deletion indicate Gnp1 again as the major serine permease in yeast. The results further demonstrate the integration of transported serine into SPs and thus the direct connection between serine uptake and SP metabolism. In addition, $[^2H_6]$-inositol labelled IPCs were slightly increased in *gnp1Δ* cells and in *gnp1Δagp1Δ* cells compared to WT cells, indicating perturbances in SP metabolism in the absence of serine uptake (Fig 6E).

In serine auxotrophic cells (*ser2Δ* cells) $[^{13}C_3^{15}N_1]$-serine and $[^2H_6]$-inositol labelled IPC levels remained in the same range as in WT cells (Fig 6D and 6E), pointing to the direct use of exogenous serine for *de novo* SP biosynthesis. If mainly serine synthesized by yeast cells would be used for SP metabolism, a decrease in labelling between WT cells and serine auxotrophic cells would be expected. Together, these results suggest a model in which Gnp1-transported serine is the main source of serine for the synthesis of SPs.

## Discussion

Here we show that external serine uptake is crucial to maintain SP homeostasis in *S. cerevisiae*. We identify the broad-specificity amino acid permease Gnp1 as the main serine transporter in

yeast. Loss of Gnp1 leads to a major reduction in serine uptake and renders the cells sensitive to chemical inhibition of SP biosynthesis. While LCB levels in *gnp1Δ* cells are relatively constant under normal growth conditions, the cells are unable to regulate SP biosynthesis according to need. Using metabolic flux analysis, we further show that *de novo* SP biosynthesis mainly depends on the uptake of extracellular serine by Gnp1 and its paralog Agp1. Together, these findings add serine uptake as another mechanism to maintain SP homeostasis in yeast.

Our data confirm previous studies that have found the overexpression of Gnp1 and Agp1 to result in increased uptake of serine [17]. Similar to previous results we detect small rates of serine uptake in the *AGP1, GNP1* double deletion strain which is potentially due to the permease Dip5 [17]. However, Dip5 does not seem to take up sufficient serine to support growth in the absence of *GNP1* and *SER2*. In addition, it has been reported that heat stress results in the upregulation of SP biosynthesis. Serine for increased SP biosynthetic rates is mainly taken from external sources [18]. This suggests that the cytosolic pools of serine are not sufficient or spatially unavailable for increased SP biosynthesis rates. Intracellular serine concentrations in yeast differ significantly across the literature [36,37]. In the experimental conditions applied herein, the internal serine concentration is in the range of 3 to 6 nmol/$10^8$ cells. Since the majority of amino acids, including serine, are sequestered in the yeast vacuole [38,39], this pool of serine would not be available for SP biosynthesis. This could be especially important under conditions where yeast has to synthesize large amounts of SPs in a short time. Thus, it is a very interesting model that serine uptake is directly coupled to *de novo* SP biosynthesis by the SPT (Fig 7).

The SPOTS complex (SPT, Orm1/2, Tsc3, and Sac1) is present in both, the nuclear as well as the cortical ER of yeast cells [7,31]. The cortical ER has recently been shown to be closely associated with the plasma membrane in ER-membrane contact sites [40,41]. Thus, Gnp1-mediated serine uptake could be directly coupled to SP biosynthesis. Besides all efforts, we were so far unable to localize a functional Gnp1 construct in cells (S7 Fig). Thus, it remains open, if Gnp1 co-localizes with the SPOTS complex of the cortical ER. However, the local generation of LCBs at ER plasma membrane contact sites has been suggested previously [42].

Amongst other mechanisms, SPT is negatively regulated by phosphorylation of the associated Orm proteins [7]. This signaling cascade starts at the plasma membrane with the re-localization of the Slm1/2 proteins to the TOR complex 2 and subsequent phosphorylation of the Ypk1 kinase, which then phosphorylates the Orm proteins [8,9]. The exact mechanism that leads to upregulation of SPT activity after Orm phosphorylation is still not clear, but recent data suggest that phosphorylated Orm2 is transported to the Golgi and degraded by the endosome-Golgi associated degradation (EGAD) pathway [31]. Since TORC2 and the Ypk kinases are located at the plasma membrane [8,10] it seems likely that downstream effectors, such as the SPOTS complex, are also located in close proximity. In this model, two pools of SPT are present in the cells i) a pool localized at the nuclear envelope that is responsible for constant SP biosynthesis and ii) a pool that can be actively regulated to synthesize large amounts of SPs according to need (Fig 7). In addition, this model could also be an explanation of the different behavior of Orm1 and Orm2. While Orm2 appears to be a target of the EGAD pathway and is mainly regulated by Ypk1, Orm1 seems to be mainly regulated via the Npr1 kinase, via the TOR complex 1 signaling pathway [9,31]. However, data obtained from our metabolic flux analysis suggest that external serine is the main source for SP biosynthesis.

In summary, we propose a model where the upregulation of the SPT is directly coupled to nutrient uptake, most likely at ER/plasma membrane contact sites (Fig 7). This system could work similar to other processes where the uptake/release of a small molecule is directly coupled to downstream pathways. One such example is local calcium dynamics at ER/mitochondria contact sites [43]. Another example where substrate availability is directly coupled to a downstream process is the recently described fatty acid channeling into phospholipid

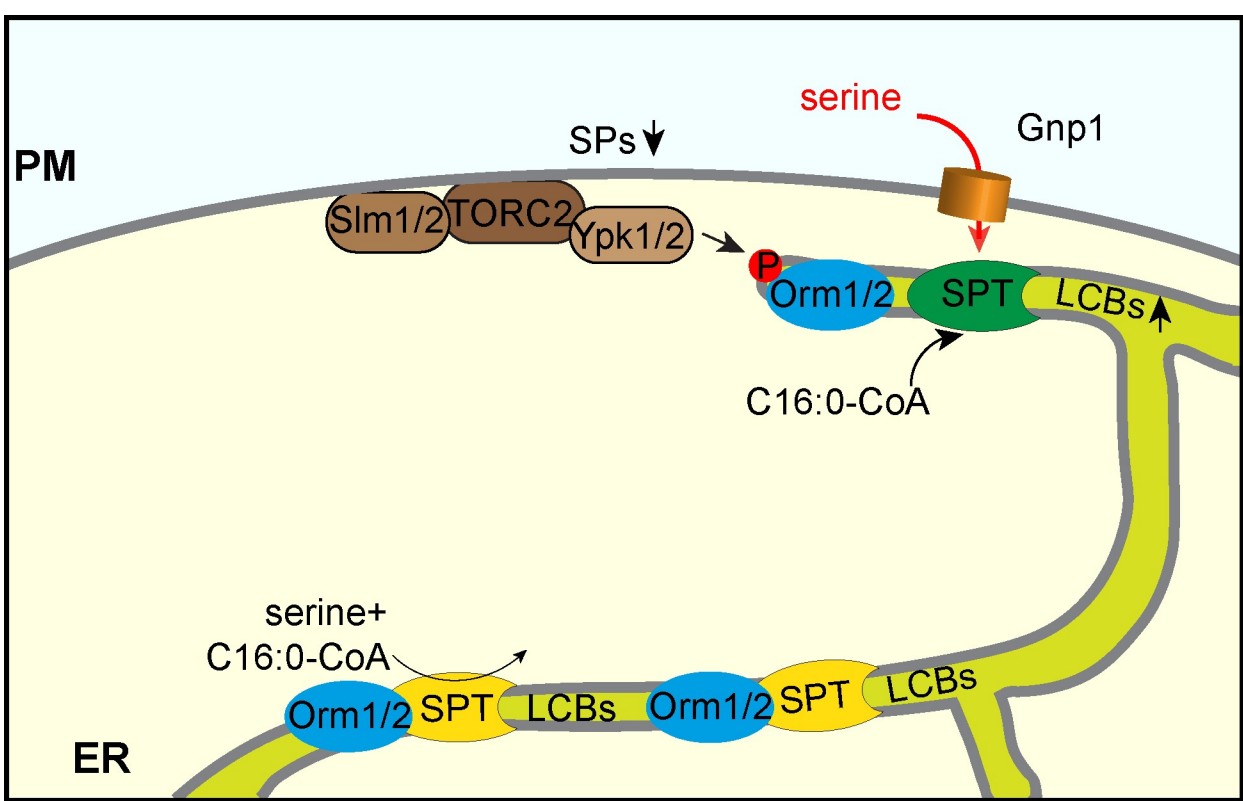

**Fig 7. Model for direct coupling of Gnp1 mediated serine uptake and regulated SP biosynthesis.** SP biosynthesis is directly coupled to Gnp1-mediated serine uptake. The SPT pool at the cortical ER is highly regulated by serine uptake and the TORC1/Ypk1/Orm1/2 signaling network. The nuclear envelope localized SPT pool is responsible for constant SP biosynthesis. This model adds serine uptake as another factor to previously described mechanisms regulating SP homeostasis by the signaling cascade of Slm1/2-TORC2-Ypk1/2 and the Orm proteins [7–9].

biosynthetic enzymes at sites of autophagosome formation [44]. In any case, our model would also require direct regulation of Gnp1. This could be achieved by manipulating the abundance of Gnp1 via regulated biosynthesis or degradation, or by regulating its transport activity by post-translational modifications. Gnp1 mRNA was shown to be regulated by the Ssy1p–Ptr3p–Ssy5 (SPS) sensor of extracellular amino acids upon the addition of leucine [45]. Moreover, we detected a modest increase in the expression of Gnp1 in response to the addition of serine to the media (S3C Fig). As an additional regulatory mechanism, we have previously identified a phosphorylation site at serine 124 of Gnp1 after myriocin treatment, indicating its regulation in response to SP depletion [14]. Moreover, Gnp1 is differentially phosphorylated in response to the TORC1 inhibitor rapamycin in *npr1Δ* cells. Npr1 has been shown to regulate plasma membrane proteins by regulated endocytosis via the arrestin proteins [46–48]. Thus, it is interesting to speculate that Gnp1 could be affected by TORC1, which has also been linked to SP homeostasis [49]. Interestingly, the genetic data presented herein also revealed synthetic phenotypes with multiple components of the TORC1 in serine auxotroph cells (S1B Fig). Testing this hypothesis will require functional fluorescent tags of Gnp1 for cell biological and biochemical approaches, whose construction failed so far (S7 Fig). Another link between TORC1 signaling, Gnp1 related serine uptake and SP metabolism comes from studies of the immunomodulating drug and sphingosine analog fingolimod (FTY720, [50]). Fingolimod was shown to modulate the activity of multiple plasma membrane permeases resulting in decreased TORC1 activity and endocytosis of plasma membrane permeases [51]. Interestingly a

mutation in Gnp1 (W239L) that causes FTY720 resistance targets a conserved amino acid in yeast plasma membrane permeases in the transmembrane helix 3 [52]. In the methionine transporter Mup1, mutation of the analog tryptophan results in increased stability of the protein in the plasma membrane [53]. Two studies have suggested that FTY720 inhibits the ceramide synthase [54,55]. Together, the increased stability of $Gnp1_{W239L}$ and elevated serine uptake could help to suppress the sphingolipid dependent phenotype of FTY720.

Varying extracellular serine concentrations have been linked to SP metabolism in mammalian cells [56,57]. Therefore, it is possible that salient features of the serine uptake-SP regulatory system are also conserved. While Gnp1 does not have a clear homolog in mammalian cells, ASCT1 has been identified as a serine transporter in brain cells [58]. Interestingly, mutations in the brain serine transporter ASCT1 (Slc1a4) have been linked to neurodevelopmental/ neurodegenerative disorders [59,60]. Although no changes in SP levels were observed in ASCT1 knockout mice so far, this could be explained by the sole determination of steady state SP levels in these studies [61]. In another neurological condition, HSNA1 (hereditary sensory and autonomic neuropathy 1), mutations in SPT result in the incorporation of alanine yielding desoxy-SPs [43]. Interestingly, high dose oral serine supplementation appears to have positive effects on patients, suggesting a functional serine uptake in their cells [62,63]. While the regulation of SP homeostasis by the ORMDL proteins in mammalian cells still remains mechanistically poorly understood, the components we identified in the yeast system are generally also present in mammalian cells. Thus, the coupling of serine uptake and SP homeostasis might be evolutionary conserved.

## Materials and methods

### Yeast strains and plasmids

Yeast strains used in this study are listed in Table 1. All plasmids used in this study are listed in Table 2. All tetrad analysis experiments were performed in the W303 strain background, except labelled otherwise. All other experiments including serine uptake assays, proteomics and lipidomics experiments were performed in the SEY6210 background. C-terminal tagging of Gnp1 and Agp1 with the ALFA-tag at their genomic locus was performed according to standard procedures [33,34].

### Yeast media and growth conditions

Tetrad dissections were performed on standard YPD plates. To generate a synthetic medium (SDC) with or without serine we used 6.7 g/l yeast nitrogen base, 20 g/l glucose and self-made drop out mix consisting of 20 mg/l adenine; 60 mg/l leucine; 20 mg/l alanine; 100 mg/l aspartic acid; 100 mg/l glutamic acid; 20 mg/l histidine; 30 mg/l isoleucine; 30 mg/l lysine; 20 mg/l methionine; 50 mg/l phenylalanine; 200 mg/l threonine; 20 mg/l tryptophan; 30 mg/l tyrosine; 20 mg/l uracil and 150 mg/l valine. To generate synthetic medium without amino acids (SD) we used 6.75 g/l yeast nitrogen base and 20 g/l glucose. Serine was added where indicated to a final concentration of 400 mg/l.

For serine incorporation measurements into proteins and measurements of free intracellular $[^{13}C_3{}^{15}N_1]$-serine levels, cells were grown in SDC medium (see dropout mix above) containing 400 mg/l $[^{13}C_3{}^{15}N_1]$-serine for 28 hours. A preculture was grown for 6 hours. Afterwards the cells were inoculated over night for 15 hours to reach $OD_{600} = 1$ in the next morning. From this culture the main culture was inoculated and grown for 7 hours to reach an $OD_{600}$ of 1.4.

For SILAC labeling, lysine auxotrophic strains SEY6210 and FFY727 were grown in SDC medium containing the same dropout mix with either 30 mg/L lysine (SEY6210) or 30 mg/L $[^{13}C_6{}^{15}N_2]$-lysine (FFY727) to $OD_{600} = 0.7$ for at least 15 generation times to ensure full labeling.

**Table 1. List of all yeast strains and their genotypes used in this study.**

| | | |
|---|---|---|
| TWY138 | Mat a *ura3-52 trp1Δ 2 leu2-3112 his3-11 ade2-1 can1-100* | Walther et al. 2007 |
| TWY139 | Mat α *ura3-52 trp1Δ 2 leu2-3112 his3-11 ade2-1 can1-100* | Walther et al. 2007 |
| FFY1260 | Mat a/α *ura3-52/ura3-52; trp1Δ 2/trp1Δ 2; leu2-3,112/leu2-3,112; his3-11/his3-11; ade2-1/ade2-1; can1-100/can1-100 SER2/ser2Δ::KAN GNP1/gnp1Δ::NAT* | this study |
| FFY1420 | Mat a/α *ura3-52/ura3-52; trp1Δ 2/trp1Δ 2; leu2-3,112/leu2-3,112; his3-11/his3-11; ade2-1/ade2-1; can1 100/can1-100 AGP1/agp1Δ::HPH GNP1/gnp1Δ::NAT SER2/ser2Δ::KAN* | this study |
| FFY1534 | Mat a/α *ura3-52/ura3-52; trp1Δ 2/trp1Δ 2; leu2-3,112/leu2-3,112; his3-11/his3-11; ade2-1/ade2-1; can1-100/can1-100 AGP1/agp1Δ::HPH SER2/ser2Δ::KAN* | this study |
| FFY2104 | Mat a/α *ura3-52/ura3-52; trp1Δ 2/trp1Δ 2; leu2-3,112/leu2-3,112; his3-11/his3-11; ade2-1/ade2-1; can1- 100/can1-100 ORM1/orm1 Δ::NAT ORM2/orm2 Δ::HPH AGP1/agp1Δ::HIS GNP1/gnp1Δ::KAN* | this study |
| FFY1901 | Mat a/α *ura3-52/ura3-52; trp1Δ 2/trp1Δ 2; leu2-3,112/leu2-3,112; his3-11/his3-11; ade2-1/ade2-1; can1-100/can1-100 SER2/ser2Δ::NAT SUR2/sur2Δ::KAN* | this study |
| FFY1902 | Mat a/α *ura3-52/ura3-52; trp1Δ 2/trp1Δ 2; leu2-3,112/leu2-3,112; his3-11/his3-11; ade2-1/ade2-1; can1-100/can1-100 GNP1/gnp1Δ::NAT SUR2/sur2Δ::KAN* | this study |
| FFY1897 | Mat a/α *ura3-52/ura3-52; trp1Δ 2/trp1Δ 2; leu2-3,112/leu2-3,112; his3-11/his3-11; ade2-1/ade2-1; can1-100/can1-100 SER2/ser2Δ::NAT SCS7/scs7Δ::KAN* | this study |
| FFY1898 | Mat a/α *ura3-52/ura3-52; trp1Δ 2/trp1Δ 2; leu2-3,112/leu2-3,112; his3-11/his3-11; ade2-1/ade2-1; can1-100/can1-100 GNP1/gnp1Δ::NAT SCS7/scs7Δ::KAN* | this study |
| FFY1899 | Mat a/α *ura3-52/ura3-52; trp1Δ 2/trp1Δ 2; leu2-3,112/leu2-3,112; his3-11/his3-11; ade2-1/ade2-1; can1-100/can1-100 SER2/ser2Δ::NAT SUR4/sur4Δ::KAN* | this study |
| FFY1900 | Mat a/α *ura3-52/ura3-52; trp1Δ 2/trp1Δ 2; leu2-3,112/leu2-3,112; his3-11/his3-11; ade2-1/ade2-1; can1-100/can1-100 GNP1/gnp1Δ::NAT SUR4/sur4Δ::KAN* | this study |
| FFY2051 | Mat a/α *ura3-52/ura3-52; trp1Δ 2/trp1Δ 2; leu2-3,112/leu2-3,112; his3-11/his3-11; ade2-1/ade2-1; can1-100/can1-100 SER2/ser2Δ::KAN LCB4/lcb4Δ::NAT* | this study |
| FFY2152 | Mat a/α *ura3-52/ura3-52; trp1Δ 2/trp1Δ 2; leu2-3,112/leu2-3,112; his3-11/his3-11; ade2-1/ade2-1; can1-100/can1-100 GNP1/gnp1Δ::KAN LCB4/lcb4Δ::NAT* | this study |
| FFY2144 | Mat a/α *ura3-52/ura3-52; trp1Δ 2/trp1Δ 2; leu2-3,112/leu2-3,112; his3-11/his3-11; ade2-1/ade2-1; can1-100/can1-100 SER2/ser2Δ::NAT LCB3/lcb3Δ::KAN* | this study |
| FFY2146 | Mat a/α *ura3-52/ura3-52; trp1Δ 2/trp1Δ 2; leu2-3,112/leu2-3,112; his3-11/his3-11; ade2-1/ade2-1; can1-100/can1-100 GNP1/gnp1Δ::NAT LCB3/lcb3Δ::KAN* | this study |
| FFY2306 | Mat a/α *ura3-52/ura3-52; trp1Δ 2/trp1Δ 2; leu2-3,112/leu2-3,112; his3-11/his3-11; ade2-1/ade2-1; can1-100/can1-100 SER2/ser2Δ::KAN GNP1/Gnp1-mcherry::KAN* | this study |
| FFY1254 | Mat α *ura3-52 trp1Δ 2 leu2-3112 his3-11 ade2-1 can1-100 ser2Δ::KAN* | this study |
| FFY1251 | Mat a *ura3-52 trp1Δ 2 leu2-3112 his3-11 ade2-1 can1-100 gnp1Δ::NAT* | this study |
| FFY1545 | Mat a *ura3-52 trp1Δ 2 leu2-3112 his3-11 ade2-1 can1-100 agp1Δ::HPH gnp1Δ::NAT* | this study |
| FFY1533 | Mat a *ura3-52 trp1Δ 2 leu2-3112 his3-11 ade2-1 can1-100 agp1Δ::HPH* | this study |
| FFY1861 | Mat a *ura3-52 trp1Δ 2 leu2-3112 his3-11 ade2-1 can1-100 orm1Δ::NAT orm2Δ::HPH* | this study |
| FFY1815 | Mat a *ura3-52 trp1Δ 2 leu2-3112 his3-11 ade2-1 can1-100 gnp1Δ::KAN* | this study |
| FFY1058 | Mat a *ura3-52 trp1Δ 2 leu2-3112 his3-11 ade2-1 can1-100 fat1Δ::NAT* | this study |
| FFY978 | Mat α *ura3-52 trp1Δ 2 leu2-3112 his3-11 ade2-1 can1-100 sur2Δ::NAT* | this study |
| FFY1829 | Mat α *ura3-52 trp1Δ 2 leu2-3112 his3-11 ade2-1 can1-100 scs7Δ::NAT* | this study |
| FFY197 | Mat α *ura3-52 trp1Δ 2 leu2-3112 his3-11 ade2-1 can1-100 sur4Δ::NAT* | this study |
| FFY1978 | Mat α *ura3-52 trp1Δ 2 leu2-3112 his3-11 ade2-1 can1-100 lcb4Δ::KAN* | this study |
| FFY2102 | Mat α *ura3-52 trp1Δ 2 leu2-3112 his3-11 ade2-1 can1-100 lcb3Δ::NAT* | this study |
| FFY1816 | Mat α *ura3-52 trp1Δ 2 leu2-3112 his3-11 ade2-1 can1-100 ser2Δ::NAT* | this study |
| FFY2439 | *ura3Δ0; leu2Δ0; his3Δ1; met15Δ0 ser2Δ::NAT* | this study |
| FFY1422 | Mat α *ura3-52 trp1Δ 2 leu2-3112 his3-11 ade2-1 can1-100 Gnp1-mcherry::KAN* | this study |
| FFY2577 | *his3Δ 1/his3Δ 1 leu2Δ 0/ leu2Δ 0 lys2Δ 0/ lys2Δ 0 ura3Δ/ura3Δ GFP-Gnp1 SER2/ser2Δ::NAT* | this study |

*(Continued)*

**Table 1.** (Continued)

| | | |
|---|---|---|
| | *MATα his3Δ1 leu2Δ0 lys2Δ0 ura3Δ0 GFP-GNP1* | [81] |
| SEY6210 | Mat α *leu2-3,112 ura3-52 his3-Δ200 trp-Δ901 lys2-801 suc2-Δ9 GAL* | Robinson et al. 1988 |
| FFY2931 | Mat α *leu2-3,112 ura3-52 his3-Δ200 trp-Δ901 lys2-801 suc2-Δ9 GAL ser1Δ::HPH* | this study |
| FFY1379 | Mat α *leu2-3,112 ura3-52 his3-Δ200 trp-Δ901 lys2-801 suc2-Δ9 GAL ser2Δ::KAN* | this study |
| FFY2932 | Mat α *leu2-3,112 ura3-52 his3-Δ200 trp-Δ901 lys2-801 suc2-Δ9 GAL ser3Δ::HPH* | this study |
| FFY2933 | Mat α *leu2-3,112 ura3-52 his3-Δ200 trp-Δ901 lys2-801 suc2-Δ9 GAL ser33Δ::HPH* | this study |
| FFY727 | Mat α *leu2-3,112 ura3-52 his3-Δ200 trp1-Δ901 suc2-Δ9 lys2-801; GAL gnp1Δ::NAT* | this study |
| FFY1375 | Mat α *leu2-3,112 ura3-52 his3-Δ200 trp-Δ901 lys2-801 suc2-Δ9 GAL agp1Δ::KAN* | this study |
| FFY1385 | Mat α *leu2-3,112 ura3-52 his3-Δ200 trp-Δ901 lys2-801 suc2-Δ9 GAL gnp1Δ::NAT agp1Δ::HPH* | this study |
| FFY2337 | Mat α *leu2-3,112 ura3-52 his3-Δ200 trp-Δ901 lys2-801 suc2-Δ9 GAL pdr5Δ::NAT* | this study |
| FFY2338 | Mat α *leu2-3,112 ura3-52 his3-Δ200 trp-Δ901 lys2-801 suc2-Δ9 GAL pdr5Δ::HPH* | this study |
| FFY2150 | Mat α *leu2-3,112 ura3-52 his3-Δ200 trp-Δ901 lys2-801 suc2-Δ9 GAL* pHLUK pRS414 | this study |
| FFY2151 | Mat α *leu2-3,112 ura3-52 his3-Δ200 trp1-Δ901 suc2-Δ9 lys2-801; GAL gnp1Δ::NAT* pHLUK pRS414 | this study |
| FFY2152 | Mat α *leu2-3,112 ura3-52 his3-Δ200 trp-Δ901 lys2-801 suc2-Δ9 GAL agp1Δ::KAN* pHLUK pRS414 | this study |
| FFY2153 | Mat α *leu2-3,112 ura3-52 his3-Δ200 trp-Δ901 lys2-801 suc2-Δ9 GAL gnp1Δ::NAT agp1Δ::HPH* pHLUK pRS414 | this study |
| FFY2214 | Mat α *leu2-3,112 ura3-52 his3-Δ200 trp-Δ901 lys2-801 suc2-Δ9 GAL ser2Δ::KAN* pHLUK pRS414 | this study |
| FFY743 | *leu2-3,112 ura3-52 his3-Δ200 trp1-Δ901 ade2-101 suc2-Δ9; GAL ser2Δ::KAN* | this study |
| FFY2984 | *leu2-3,112/leu2-3,112 ura3-52/ura3-52 his3-Δ200/his3-Δ200 trp1-Δ901/trp1-Δ901 ade2/ADE2 suc2-Δ9/suc2-Δ9 GAL/GAL LYS2/lys2-801 gnp1Δ::NAT ser2Δ::KAN* | this study |
| FFY2763 | *leu2-3,112/leu2-3,112 ura3-52/ura3-52 his3-Δ200/his3-Δ200 trp1-Δ901/trp1-Δ901 ade2/ADE2 suc2-Δ9/suc2-Δ9 GAL/GAL LYS2/lys2-801 Gnp1-ALFA::HIS ser2Δ::KAN* | this study |
| FFY2721 | *leu2-3,112 ura3-52 his3-Δ200 trp1-Δ901 ade2-101 suc2-Δ9; GAL Gnp1-ALFA::HIS* | this study |
| FFY2934 | *leu2-3,112 ura3-52 his3-Δ200 trp1-Δ901 ade2-101 suc2-Δ9; GAL agp1Δ::KAN Gnp1-ALFA::HIS* | this study |
| FFY2935 | *leu2-3,112 ura3-52 his3-Δ200 trp1-Δ901 ade2-101 suc2-Δ9; GAL Agp1-ALFA::HIS* | this study |
| FFY2936 | *leu2-3,112 ura3-52 his3-Δ200 trp1-Δ901 ade2-101 suc2-Δ9; GAL gnp1Δ::NAT Agp1-ALFA::HIS* | this study |
| FFY2939 | *leu2-3,112 ura3-52 his3-Δ200 trp1-Δ901 ade2-101 suc2-Δ9; GAL Gnp1-ALFA::HIS* pHLUK pRS414 | this study |
| FFY2940 | *leu2-3,112 ura3-52 his3-Δ200 trp1-Δ901 ade2-101 suc2-Δ9; GAL agp1Δ::KAN Gnp1-ALFA::HIS* pHLUK pRS414 | this study |
| FFY2937 | *leu2-3,112 ura3-52 his3-Δ200 trp1-Δ901 ade2-101 suc2-Δ9; GAL Agp1-ALFA::HIS* pHLUK pRS414 | this study |
| FFY2938 | *leu2-3,112 ura3-52 his3-Δ200 trp1-Δ901 ade2-101 suc2-Δ9; GAL agp1Δ::KAN Gnp1-ALFA::HIS* pHLUK pRS414 | this study |

## Spotting assays

For spotting assays, cells were grown to exponential growth phase and serial diluted. Cells were spotted on the corresponding plates and incubated at 30 ˚C for three days. YPD and SD media with and without amino acids were used as indicated. Chemicals were added in the indicated concentrations.

**Table 2. List of all plasmids used in this study.**

| pHLUK | ADDGENE |
|-------|---------|
| pRS414 | [82] |

## Quantification of colony sizes

ImageJ was used for the quantification of colony sizes of single tetrads. The obtained picture was converted as a binary. A circle was placed around each tetrad and mean intensity values were measured.

## Serine uptake assays

Amino acid uptake assays were performed as previously described [64] with minor modifications. Briefly, WT, *gnp1Δ*, *agp1Δ* and *gnp1Δagp1Δ* cells were grown in synthetic medium with serine for at least 13 doublings to reach an $OD_{600}$ = 0.5. With the aid of a vacuum system, cells were filtered and resuspended from the filters (MF Membrane Filters 0.45 μm, Merck KGaA, 64293 Darmstadt, Germany) in the same volume of synthetic medium without serine. $OD_{600}$ was measured and 0.72 mM [$^{14}$C]-serine were added. At 5, 15 and 30 min 1 ml of the culture was filtered with serine-saturated filters. To measure the background 1 ml of the 0.72 mM [$^{14}$C]-serine solution was also filtered the same way. The filters were covered by a scintillation liquid and the radioactivity was measured with a scintillation counter (Beckman Coulter LS 6500 Multi-Purpose Scintillation Counter, GMI Trusted Laboratory Solutions, MN 55303, USA). The transport was expressed as nmol/mg of protein per unit of time and reported as the mean ± S.D. ($n$ = 3).

## Proteomics

Proteomics analysis was performed as described previously [65]. Briefly, yeast cells were lysed using the filter aided sample preparation method [66]. For full proteome analysis, samples were eluted from a PepMap C18 easy spray column (Thermo) with a linear gradient of acetonitrile from 10–35% in $H_2O$ with 0.1% formic acid for 118 min at a constant flow rate of 300 nl/min.

For serine incorporation assays using [$^{13}C_3^{15}N_1$]-serine (Cambridge Isotope Labs), cells were lysed and peptides were cleaned up using the iST kit (Preomics) and digested with the α-Lytic protease (New England BioLabs). Peptides were eluted from a PepMap C18 easy spray column (Thermo) with a linear gradient of acetonitrile from 10–50% in 0.1% formic acid for 33 min at a constant flow rate of 300 nl/min.

The resulting MS and MS/MS spectra were analyzed using MaxQuant (version 1.6.0.13, www.maxquant.org/ [67,68]) as described previously [65]. For incorporation tests, [$^{13}C_3^{15}N_1$]-serine was added as a modification to the MaxQuant database (mass change = 4.0070994066 Da). For the calculation of incorporation rates, the peptide list was filtered for serine containing peptides with a valid heavy/light ratio. For each peptide, the incorporation was calculated as 1 - (1/(ratio H/L—1)). The maximum of a density distribution of all peptides represents the estimated incorporation level. All calculations and plots were performed with the R software package (www.r-project.org/; RRID:SCR_001905).

## LCB analysis

For LC-MS analysis of LCBs, cells were grown in YPD to exponential growth phase and treated for 3 hours with the indicated concentrations of myriocin. Lipids were extracted

from lysed yeast cells according to 50 μg of protein by chloroform/methanol extraction [69]. Prior to extraction sphingosine (LCB 17:0) was spiked into each sample for normalization and quantification. Dried lipid samples were dissolved in a 65:35 mixture of mobile phase A (60:40 water/acetonitrile, including 10 mM ammonium formate and 0.1% formic acid) and mobile phase B (88:10:2 2-propanol/acetonitrile/$H_2O$, including 2 mM ammonium formate and 0.02% formic acid). HPLC analysis was performed on a C30 reverse-phase column (Thermo Acclaim C30, 2.1 × 250 mm, 3 μm, operated at 50 ˚C; Thermo Fisher Scientific) connected to an HP 1100 series HPLC (Agilent) HPLC system and a QEx-active*PLUS* Orbitrap mass spectrometer (Thermo Fisher Scientific) equipped with a heated electrospray ionization (HESI) probe. The analysis was performed as described previously [70]. Peaks were analyzed using the Lipid Search algorithm (MKI, Tokyo, Japan). Peaks were defined through raw files, product ion and precursor ion accurate masses. Candidate molecular species were identified by database (>1,000,000 entries) search of positive ($+H^+$) ion adducts. Mass tolerance was set to five ppm for the precursor mass. Samples were aligned within a time window 0.5 min and results combined in a single report. From the intensities of the lipid standard absolute values for each lipid in pmol/mg protein were calculated. Data are displayed as fold change from WT.

## Quantification of cellular amino acids

Prototroph strains were grown in synthetic medium without amino acids (except of the addition of serine were indicated), collected by centrifugation and washed with ice-cold $H_2O$ for three times. Cells for the measurement of free intracellular $[^{13}C_3{}^{15}N_1]$-serine were grown as described in the section "Yeast media and growth conditions", collected by centrifugation and washed with ice-cold $H_2O$ for three times. Cell lysis and amino acid extractions were performed following the modified protocol from [37]. Ethanol containing the non-proteinogenic amino acid norleucin was added to the pellet, mixed and heated to 80˚ C for 5 minutes. The extract was sonicated for 15 minutes and the heating step was repeated. The extract was cleared by centrifugation, dried and resolved in $H_2O$ with 10% acetonitrile. The amino acids were derivatized with dansyl-chloride and analysed by LC-MS/MS.

Amino acids were separated by hydrophilic interaction liquid chromatography using a Shimadzu Nexera HPLC with ThermoFisher Accurore RP-MS C18 column (0.21 x 150 mm; 2.6 μm particle size). For the mobile phases 0.1% formic acid in water (A) and 0.1% formic acid in 80% acetonitrile (B) was used. For the gradient 0 to 50% B over 0.7 min, 50% to 60% over 3.7 min, 60% to 100% B over 0.1 min, 100% B kept for 0.5min and 0% B for 1.5 min and a constant flow rate of 0.4 ml/min was used with a total analysis time of 6 minutes and an injection volume of 1 μl. The samples were analyzed by a QTRAP 5500 LC-MS/MS system (SCIEX), with IonDrive TurboV source. The MS data were acquired in positive, scheduled MRM mode with 20 sec detection windows (Gly: Q1 309, Q3 170, RT 2.65min, DP 111V, CE, 27V, CXP 26V; Ser: Q1 339, Q3 170, RT 2.02min, DP 36V, CE, 29V, CXP 6V; Lys: Q1 380, Q3 170, RT 1.73min, DP 116V, CE, 43V, CXP 6V; $[^{13}C_3{}^{15}N_1]$-Ser: Q1 349, Q3 170, RT 2.27min, DP 36V, CE 29V, CXP 6V). For peak integration SciexOS software was used. The Amino Acid Standard H from Thermo Fischer was used to quantify the amino acids. The amino acid concentrations were expressed in nmol per $10^8$ cells and reported as the mean ± S.D. ($n$ = 3).

## Modelling flux variability

Flux variability analysis (FVA) [71] was performed on the yeast consensus genome-scale model yeast-GEM-v7.6.0 [72] and constraints were adjusted to account for the gene deletions

in the SEY6210 mutant and to simulate growth in SDC medium or to simulate growth in YPD medium by adding the uptake-fluxes of all 20 amino acids and 4 nucleotide bases [73]. The computed minimum and maximum fluxes correspond to the solution space which supports 99% of maximum feasible biomass growth. We used COBRA Toolbox v3.0 [74] in MATLAB (The MathWorks, Inc.) and employed the algorithm fastFVA [75] and CPLEX 12.8 (IBM Corp.) to solve optimization problems. The metabolic model is available at https://github.com/SysBioChalmers/yeast-GEM/tree/v7.6.0. Our MATLAB script is provided in S1 Script for simulations in SDC medium and S2 Script for simulations in YPD medium.

## Western blotting

Cells were grown in the indicated growth medium. Proteins were precipitated with TCA and analyzed via western blotting. ALFA-tagged proteins were detected with a mouse anti-ALFA (HRP) antibody (NanoTag; N1502-HRP) diluted 1:1000. Pgk1 was detected with an anti-Pgk1 antibody diluted 1:5000 (Thermo; RRID:AB_2532235) using a horseradish peroxidase coupled mouse IgG kappa binding protein (Santa Cruz biotechnology; RRID:AB_2687626).

## Lipidomics flux analysis

Yeast strains were pre-cultured in YPD. At an $OD_{600}$ of 0.8, 5 ml cell suspensions were spiked with a tracer cocktail resulting in final concentrations of 4.59 mM $[^{13}C_3{}^{15}N_1]$-serine and 0.22 mM $[^2H_6]$-inositol. After 90 minutes, the cell suspensions were incubated at 4 ˚C for 10 min with 1M perchloric acid. The cells were centrifuged and washed with 4 ˚C 155 mM ammonium formate, frozen in liquid nitrogen and stored at –80 ˚C.

In-depth lipidome analysis was performed as described previously [69,76]. In short, yeast cell pellets (∼5 ODunits) were resuspended in 1 ml of 155 mM ammonium formate and lysed at 4 ˚C with 400 μl of acid-washed glass beads using a Mini-Beadbeater (Biospec). Lysates corresponding with 0.4 OD units were subsequently spiked with internal lipid standards and subjected to two-step lipid extraction [69]. Lipid extracts were vacuum evaporated and re-dissolved in 100 μl chloroform/methanol (1:2; v/v) and analyzed by $MS^{ALL}$ [77] using an Orbitrap Fusion Tribrid (Thermo Fisher Scientific) equipped with a TriVersa NanoMate robotic nanoflow ion source (Advion Biosciences). Lipid identification and quantification was done using ALEX$^{123}$ software [78–80]. The results were expressed in mol % per all detected lipids ($n$ = 2).

## Supporting information

**S1 Fig. (a)** The genetic interaction score (epsilon score) of *SER1* is plotted against the negative $LOG_{10}$ of the *p*-value of the interactions. The volcano plot shows significant negative genetic interactions on the left side of the plot. Data are taken from [24]. **(b)** The genetic interaction score (epsilon score) of *SER2* is plotted against the negative $LOG_{10}$ of the *p*-value of the interactions. The volcano plot shows significant negative genetic interactions on the left side of the plot. Dots are color coded according to the respective signaling pathways (orange–TORC1; read–SEA complex, purple–EGO complex, green–unknown signaling pathway). Data are taken from [24].
(TIF)

**S2 Fig. (a)** Predicted serine hydroxymethyltransferases (Shm) net fluxes. Variability of net flux through Shm1 and Shm2 at varying serine uptake rates, as predicted by FVA. Positive and negative fluxes correspond to net production of serine and glycine, respectively. Fluxes and serine uptake rates are represented in mmol per gram dry weight per hour. **(b)** Cellular lysine concentrations. Prototroph WT and gnp1Δ cells were grown in synthetic media without

amino acids and with and without serine. Lysine concentrations from whole cell lysates were analyzed by mass spectrometry. Error bars represent standard deviations. n = 3.
(TIF)

**S3 Fig. (a)** Tetrad analysis of SEY6211 *ser2Δ* (blue) mutants crossed with SEY6210 *gnp1Δ* cells (red). **(b)** Tetrad analysis of SEY6211 *ser2Δ* (blue) mutants crossed with SEY6210 Gnp1-ALFA cells (red). **(c)** Expression level of Gnp1-ALFA and Agp1-ALFA. Cells were grown in YPD, SDC medium, SD medium without amino acids (AA) and SD media with serine. Equal amounts of cells were lysed and analyzed by western blotting using antibodies against the ALFA-tag or Pgk1 as a loading control. A wildtype strain was used as a control.
(TIF)

**S4 Fig. (a)** Tetrad analysis of *gnp1Δ* (red) mutants crossed with *sur2Δ* (orange). **(b)** Tetrad analysis of *ser2Δ* (blue) mutants crossed with *sur2Δ* (orange). **(c)** Tetrad analysis of *gnp1Δ* (red) mutants crossed with *scs7Δ* (orange). **d)** Tetrad analysis of *ser2Δ* (blue) mutants crossed with *scs7Δ* (orange). **(e)** Tetrad analysis of *gnp1Δ* (red) mutants crossed with *sur4Δ* (orange). **(f)** Tetrad analysis of *ser2Δ* (blue) mutants crossed with *sur4Δ* (orange). **(g)** Tetrad analysis of *gnp1Δ* (red) mutants crossed with *lcb4Δ* (orange). **(h)** Tetrad analysis of *ser2Δ* (blue) mutants crossed with *lcb4Δ* (orange). **(i)** Tetrad analysis of *gnp1Δ* (red) mutants crossed with *lcb3Δ* (orange). **(j)** Tetrad analysis of *ser2Δ* (blue) mutants crossed with *lcb3Δ* (orange).
(TIF)

**S5 Fig. (a)** Serial dilutions of WT, *gnp1Δ* cells, *agp1Δ* cells, *gnp1Δ agp1Δ* cells and two different clones of *pdr5Δ* cells on YPD plates. Control plates (left) and plates containing 1.8 μM cerulenin (right) were used. **(b)** Serial dilutions of WT, *gnp1Δ* cells, *agp1Δ* cells, *gnp1Δ agp1Δ* cells and *fat1Δ* cells on YPD plates. Control plates (left) and plates containing 0.075 μM Aureobasidin A (right) were used.
(TIF)

**S6 Fig. (a)** Integration of $[^{13}C_3{}^{15}N_1]$-serine into ceramides. Cells were labelled with $[^{13}C_3{}^{15}N_1]$-serine and $[^2H_6]$-inositol over 90 minutes in YPD media. Lipids were extracted and analyzed via mass spectrometry. Displayed are the amounts of $[^{13}C_3{}^{15}N_1]$-serine labelled ceramides of WT cells, *gnp1Δ* cells, *gnp1Δagp1Δ* cells and *ser2Δ* cells in mol% per all detected lipids. The average is displayed in bars. Dots correspond to the values of two independent experiments.
(TIF)

**S7 Fig. (a)** Tetrad analysis of *ser2Δ* (blue) mutants crossed with Gnp1-mcherry (red). **(b)** Tetrad analysis of BY *ser2Δ* (blue) mutants crossed with BY GFP-Gnp1.
(TIF)

**S1 Data. Data set for genetic interactions of SER1 and SER2 in Figs 2B and S1.** Data taken from [24].
(XLS)

**S2 Data. Data set for $[^{14}C]$-serine uptake measurements in Fig 3A.**
(XLSX)

**S3 Data. Data set for incorporation of $[^{13}C_3{}^{15}N_1]$-serine into the proteome in Fig 3C.**
(XLSX)

**S4 Data. Data set of free intracellular $[^{13}C_3{}^{15}N_1]$-serine levels in Fig 3D.**
(XLSX)

**S5 Data. List of all proteins identified including SILAC ratios and intensities from the comparison of WT and *gnp1Δ* cells in Fig 3E.**
(XLSX)

**S6 Data. Data set of serine, glycine and lysine levels of WT and *gnp1Δ* cells with and without serine presented in Figs 4C and S5B.**
(XLSX)

**S7 Data. Source data for the quantification of colony sizes of the tetrad analysis in Fig 5D and 5E.**
(XLSX)

**S8 Data. Data set of long chain base levels of WT and *gnp1Δ* cells with and without myriocin presented in Fig 6B.**
(XLSX)

**S9 Data. Data set of serine (6d) and inositol (6e) labelled IPCs and ceramides presented in Fig 6B, 6D and 6E and S6 Fig.**
(XLSX)

**S1 Script. MATLAB script used for the flux variability analysis in SDC media.**
(M)

**S2 Script. MATLAB script used for the flux variability analysis in YPD media.**
(M)

## Acknowledgments

We thank members of the Fröhlich lab for critical comments on the manuscript.

## Author Contributions

**Conceptualization:** Bianca M. Esch, Florian Fröhlich.

**Data curation:** Florian Fröhlich.

**Formal analysis:** Bianca M. Esch, Florian Fröhlich.

**Investigation:** Bianca M. Esch, Sergej Limar, André Bogdanowski, Christos Gournas, Tushar More, Celine Sundag, Stefan Walter, Jürgen J. Heinisch, Florian Fröhlich.

**Methodology:** Christer S. Ejsing, Bruno André, Florian Fröhlich.

**Project administration:** Florian Fröhlich.

**Resources:** Florian Fröhlich.

**Software:** André Bogdanowski.

**Supervision:** Florian Fröhlich.

**Validation:** Florian Fröhlich.

**Visualization:** Bianca M. Esch, Florian Fröhlich.

**Writing – original draft:** Bianca M. Esch, Florian Fröhlich.

**Writing – review & editing:** Bianca M. Esch, Florian Fröhlich.

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
