## [Decision Letter · Decision Letter 0]

18 May 2020

Dear Dr Fröhlich,

Thank you very much for submitting your Research Article entitled 'Uptake of exogenous serine is important to maintain sphingolipid homeostasis in Saccharomyces cerevisiae' to PLOS Genetics. Your manuscript was fully evaluated at the editorial level and by independent peer reviewers. The reviewers appreciated the attention to an important topic but identified some aspects of the manuscript that should be improved.

We therefore ask you to modify the manuscript according to the review recommendations before we can consider your manuscript for acceptance. Your revisions should address the specific points made by each reviewer.

[LINK]

Yours sincerely,

Aimee M. Dudley, Ph.D.

Associate Editor

PLOS Genetics

Gregory P. Copenhaver

Editor-in-Chief

PLOS Genetics

The reviewers appreciated the importance and rigor of the work. However, they collectively identified several places in the manuscript where there were issues around data interpretation or the strength of the conclusions needed to be strengthened by additional controls or replicates. We would be happy to consider a revised manuscript that addresses these points in some way.

Reviewer's Responses to Questions

**Comments to the Authors:**

Reviewer #1: This is a solid paper in which the authors clearly demonstrate that serine uptake into yeast cells, especially by the aa transporter Gnp1, supports sphingolipid (SL) biosynthesis. The authors confirm this point by several approaches (uptake assays, genetic analysis, drug sensitivity, data mining of genetic interactions, metabolic labelling and mass spectrometry, biochemical assays) and model-driven hypotheses. The article is very clear and nicely written.

In terms of originality, the paper may be a bit weaker because as the authors mention in the text, it was previously suggested that Gnp1 allows Ser uptake, but that was based on overexpression. Thus, one would expect that SL biosynthesis (of which Ser is a precursor) relies at least partly on this. Yet, the more direct demonstration of Gnp1 and Agp1 as important contributors of Ser uptake and the preferential use of exogenous Ser for SL biosynthesis is important and interesting.

My main criticism is that I do not agree with the author’s conclusion that the growth defect of the orm1D orm2D double mutant “was partly restored by the additional deletion of GNP1, but not that of AGP1” (l. 310, refers to fig 5D). First, there is only one such combination (3rd tetrad, 1st spore) and I find it is not very different from the double mutant. However I agree that the quadruple mutant is partially rescued for the 3 spores displayed. Either the authors come up with a way to quantify this growth defect more precisely, on more examples of triple mutants, measuring colony size etc., or they make a small adjustment in the text and tone down on the sole importance of Gnp1 in this rescue. Anyway, depending on the readout (Ser uptake, myriocin sensitivity, SL with ser2D etc.) there are other examples in the paper where Agp1 and Gnp1 are differentially required, so that’s not an issue and it doesn’t change the main message of the study.

Then I only had a few questions or suggestions, just in case the authors want to comment on this / add something in the paper, if it seems relevant. A previous paper isolated a dominant mutant of GNP1 as being resistant to the SL-like drug FTY720 (10.1074/jbc.M213144200) and I think the author’s work might explain some of the observations that could be further discussed, especially because this drug is used as an immunosuppressant. Also, are there previously published data (genetic interactions, transcriptomes of gnp1D mutant) further supporting a role of Gnp1 in SL biology? I know it might be compensated by Agp1, but again the relative importance of Gnp1 over Agp1 was sufficient to be picked up in the ser2D screen.

Reviewer #2: Esch et al have investigated metabolic connections between serine metabolism and sphingolipid metabolism in the yeast, Saccharomyces cerevisiae. Building on their published observation that ser2 deletion cells are super-sensitive to inhibition of serine palmitoyl transferase (SPT) by myriocin, and published systematic genetic interaction screening results, the authors identify GNP1 and the related AGP1 as genes encoding serine importers, with the activity of GNP1 predominating what are the conditions examined. Mass spectrometry-based quantitations show that deletion of GNP1 and AGP1 affects serine incorporation into protein and sphingpolipid. Genetic analyses further suggest that the SPT step of sphingolipid by synthetic pathway is influenced by serine availability, a point further supported by flux variability analyses. Genetic analyses and mass spectrometry-based quantitation of sphingolipids in wild-type and gnp1D cells show directly that Gnp1 contributes to sphingolipid synthesis via import of extracellular serine. The authors propose that this reveals a bias the use of Gnp1/Agp1-imported serine by SPT and they speculate that this represents substrate channeling at ER-PM contact sites.

This is a rigorous, well executed and nicely presented study. I have no technical issues to raise. The data supporting the conclusion that Gnp1 is the major serine transporter of the cell builds on a previously published report and is definitive. The significance of the remainder of the manuscript is less clear. A key finding of the study is presented in Fig. 6b, where deletion of GNP1 is shown to exacerbate the reduction in PHS caused by low dose myriocin, from ~20% reduction in wild type cells to ~40% reduction in gnp1D cells. While this reduction does indicate that Gnp1 is necessary to sustain sphingolipid biosynthesis, the magnitude of the reduction is modest, leaving physiological importance unclear. The idea that serine may be channeled to SPT at sites of ER and PM contact is interesting though speculative at this stage. It might be useful to include in the analyses deletions of genes encoding components required to ER-PM contact sites (e.g., this encoding tethering proteins, SCS2/22 (VAP homologs).

Reviewer #3: In this study, Esch and colleagues used budding yeast as model organism to uncover the role of exogenous serine in the regulation of de novo sphingolipid (SP) biosynthesis. Esch et al start with two observations: first, the presence of exogenous serine is important for myriocin tolerance in WT cells and second, myriocin treatment is lethal for ser2Δ cells even in the presence of external serine. Myriocin inhibits SPT, the enzyme that catalyzes the condensation of serine and palmitoyl-CoA in the first step of SP biosynthesis. The authors take these primary observations further and using a set of genetic and biochemical experiments propose that (a) Gnp1 is a major serine transporter in yeast, (b) Gnp1-mediated transport of serine and its flux through the SP biosynthetic pathway is important for overcoming sensitivity to myriocin, and (c) exogenous serine is the main source for de novo SP biosynthesis. The study is well conducted and the results are for the most part clearly presented. Sometimes, the experimental design could have been better, as for example inclusion of additional mutant strains in some of the experiments shown to reach more firm conclusions, more coherence in the growth media used across the paper. However, the study is overall very compelling and provides original findings on serine transport and its connection to sphingolipid metabolism.

Major comments:

1. Including ser2Δ in the Figure 3A experiment would have told us whether serine uptake is affected in this strain. It would have nicely complemented the results presented in Figure 3C, and would have given us an idea on how inhibiting serine synthesis (in ser2Δ) affects uptake of external serine. Also, including the agp1Δ (and agp1Δgnp1Δ) strain in Figure 3C would have clarified further the relative contributions of Agp1 and Gnp1 to serine import in the context of protein synthesis.

2. Lines 208-210: The questions, raised by the authors themselves, on whether the intracellular serine pool is in majority derived from de novo synthesis or from imported serine and to what degree the imported serine is channeled towards protein synthesis could have been addressed by measurement of the intracellular (non-protein bound) serine pool and determination of the labelled fraction by LC-MS analysis. The suggested experiment along with the results shown in Figure 6D could give additional weightage to the suggestion that imported serine is to a large extent utilized for SP biosynthesis.

3. Lines 214-217: It would be important to remind in this context that the serine auxotrophy conferred by inactivation of the 3-phosphoglycerate-dependent serine synthesis pathway in yeast is conditional, since the secondary pathway leading to serine synthesis via glycine is glucose repressed. The authors should remind this and reflect on whether, in their experimental conditions, the secondary pathway may be active or not (considering their cultivation conditions and sampling time).

4. Lines 284-285: it is difficult to understand why the absence of amino acids in the media abolishes the myriocin phenotype in the gnp1Δ strain (it behaves like wt). Some even speculative explanations by the authors would be helpful here.

5. Lines 293-298: the authors describe that Gnp1 and Agp1 expression levels could not yet be determined at the protein level. Although less ideal, why did they not at least validate their hypothesis concerning GNP1 and AGP1 expression changes in different media using qRT-PCR? As one reason for some unexpected results in Fig5 would be differential regulation of GNP1 and AGP1 expression in media with and without amino acids, these quantitative gene expression analyses should be performed in the relevant growth conditions. It would also be interesting to evaluate how the deletion of GNP1 affects AGP1 expression and vice-versa.

6. It is interesting that the authors include flux variability analyses in the study, but their relevance for the paper did not come out very clearly for this reviewer (the predictions seemed often not very surprising). It is difficult to understand that experiments conducted to test predictions of the modeling were performed in different growth conditions as the one used for doing the simulations. This is for example the case for the FVA done to predict the effect of serine uptake rate on SPT flux (Fig. 6a). Why did the authors perform the experiment to test the effect of myriocin exposure and GNP1 deletion on PHS levels in YPD (Fig. 6b), when in the modeling constraints were adjusted to growth in SDC medium?

7. In Fig. 5b, authors showed that the gnp1Δ strain has strong growth defects in YPD containing 0.5 μM myriocin, that are rescued by PHS supplementation. This strongly supports that under these conditions, PHS levels in the myriocin exposed gnp1Δ strain are limiting growth. However, in Fig. 6b, there is no significant difference in PHS levels between wt and gnp1Δ cells. The authors should comment on this apparent discrepancy and give possible explanations. As strains with three different genetic backgrounds (BY, W303, SEY6210) were used in this study and medium composition (solid/liquid, rich/synthetic) was also often changed from one experiment to another, it is not easy for the reader to disentangle possible reasons for apparent contradictions between observations. Background effects? Growth defects are always shown in spotting assays. Is the growth effect of myriocin also observed in a liquid cultivation setting? In general, efforts should be made to indicate as clearly as possible in Figure legends etc, which growth medium was used and what is the genetic background of the tested strains.

8. There also seems to be a contradiction between lines 184-186 (complete lack of serine import in the absence of Agp1 and Gnp1) and lines 355-357 (decrease of 73% labeling in SPs in agp1Δgnp1Δ cells in the presence of extracellular labeled serine). The observation that there is residual labeling (27%) of SPs in the absence of both Agp1 and Gnp1 indicates that there is an additional uptake system for extracellular serine? This is also supported by the observation that addition of serine in experiment 5c seems to rescue the growth defect caused by myriocin in gnp1Δagp1Δ cells, suggesting again that there are other serine uptake mechanisms in the yeast cell that can partially overcome the absence of both transporters. These apparent contradictions should be explicitly addressed and discussed.

Minor comments:

1. In Fig. 1d, a complete growth inhibition of the ser2Δ strain is shown in the presence of myriocin. Was the same effect observed in ser1Δ cells or is this phenotype specific for ser2Δ? As the deletion strains used were from the knockout collection, it would be preferable to show rescue experiments or make the observations more robust by using several mutants with supposedly the same metabolic consequence.

2. What is the cutoff for “highly significant interactions” in Fig. 2b? Can the authors provide the list with the most significant genetic interactors for SER2? In addition, the genetic interaction results obtained for the SER1 deletion should also be provided in supplementals.

3. In Fig. 3c, authors conclude that the ser2Δ strain is still able to synthesize serine from glycine. However, there may be remaining non-labeled serine from the previous medium? Single colonies were picked and inoculated in SDC containing 400 mg/l [13C315N1]-serine or was there a pre-culture before? In the method, it is described that cells were grown for at least 15 doublings? This is an estimation or it was indeed calculated? It would be preferable to describe the cultivation method for the serine incorporation measurements in more detail.

4. Can the authors provide a list with the functional annotations of the most significant proteins obtained in Fig. 3d? The full source data set is provided in the supplemental, but it would be useful to also provide a small list as supplemental Table. It is quite interesting for instance that ZRT1 and MZM1 (two genes involved in zinc homeostasis) are downregulated in the gnp1Δ strain. Connection between zinc and SPs is not the scope of this study but it might be useful for some readers or for future follow-up studies. In general, the authors could expand a little further on discussing the results in this figure as is currently the case.

5. In Fig. 4c, it would have been good to also show the levels of a more unrelated amino acid as control to confirm that the increase of glycine is a specific consequence of serine supplementation and not just part of a more general dysregulation of the amino acid pool by serine excess.

6. Line 277: replace 5c by 5b.

7. Line 511: “labelled for … with … myriocin.” Please reformulate.

8. Authors should not switch between ‘LCB’ and ‘PHS’ abbreviations, unless needed.

**Have all data underlying the figures and results presented in the manuscript been provided?**

Reviewer #1: Yes

Reviewer #2: Yes

Reviewer #3: Yes

PLOS authors have the option to publish the peer review history of their article (what does this mean?). If published, this will include your full peer review and any attached files.

Reviewer #1: No

Reviewer #2: No

Reviewer #3: No

---

## [Decision Letter · Decision Letter 1]

22 Jul 2020

Dear Dr Fröhlich,

We are pleased to inform you that your manuscript entitled "Uptake of exogenous serine is important to maintain sphingolipid homeostasis in Saccharomyces cerevisiae" has been editorially accepted for publication in PLOS Genetics. Congratulations!

All three reviewers appreciated your effort in revising the manuscript and felt that it was acceptable for publication in PLoS Genetics. Please note that reviewer 3 identified three very minor edits (see below) that should be made as you prepare your final draft for the production team (the editorial team will not need to revaluate).

Yours sincerely,

Aimee M. Dudley, Ph.D.

Associate Editor

PLOS Genetics

Gregory Copenhaver

Editor-in-Chief

PLOS Genetics

Comments from the reviewers (if applicable):

Reviewer's Responses to Questions

Comments to the Authors:

Please note here if the review is uploaded as an attachment.

Reviewer #1: I would like to thank the authors for their consideration - they have addressed all of my concerns.

Reviewer #2: In my opinion, the authors have satisfied all of the critical concerns and the manuscript should be published without delay.

Reviewer #3: The authors made considerable efforts to address the reviewers’ comments and all of my points were addressed thoroughly.

Remaining revision requests:

-One line 234 in the revised manuscript with tracked changes, the authors mention the growth of a gnp1 agp1 ser2 triple KO mutant. Please refer back to Fig. 2e here.

-The authors should revise the labeling of their source data files as there seems to be some mislabeling (e.g. S5 and S6)

-It is very nice that an expression analysis is now shown for Gnp1 and Agp1 in different media using ALFA tagging (FigS9c). There is a strain list, but it would be more straightforward if the authors could provide a description of how the Gnp1-ALFA and Agp1-ALFA strains were created (plasmid based expression or integration? promoter? N-terminal or C-terminal tagging? etc). In Fig S9b, was the parental strain used a rescue strain (gnp1 deletion strain in which Gnp1-ALFA is expressed)? or was Gnp1 tagged directly in its genomic locus?. This information is important to appreciate the relevance of the ALFA protein expression studies.

Have all data underlying the figures and results presented in the manuscript been provided?

Large-scale datasets should be made available via a public repository as described in the 

PLOS Genetics

data availability policy, and numerical data that underlies graphs or summary statistics should be provided in spreadsheet form as supporting information.

Reviewer #1: Yes

Reviewer #2: Yes

Reviewer #3: None

PLOS authors have the option to publish the peer review history of their article (what does this mean?). If published, this will include your full peer review and any attached files.

Do you want your identity to be public for this peer review?

 For information about this choice, including consent withdrawal, please see our Privacy Policy.

Reviewer #1: No

Reviewer #2: No

Reviewer #3: No

**Data Deposition**

http://datadryad.org/submit?journalID=pgenetics&manu=PGENETICS-D-20-00461R1

Press Queries

---

## [Editor Report · Acceptance letter]

19 Aug 2020

PGENETICS-D-20-00461R1 

Uptake of exogenous serine is important to maintain sphingolipid homeostasis in Saccharomyces cerevisiae 

Dear Dr Fröhlich, 

We are pleased to inform you that your manuscript entitled "Uptake of exogenous serine is important to maintain sphingolipid homeostasis in Saccharomyces cerevisiae" has been formally accepted for publication in PLOS Genetics! Your manuscript is now with our production department and you will be notified of the publication date in due course.

With kind regards,

Matt Lyles

PLOS Genetics

On behalf of:
